# Wireless Power Transfer: Systems, Circuits, Standards, and Use Cases

**DOI:** 10.3390/s22155573

**Published:** 2022-07-26

**Authors:** Jarne Van Mulders, Daan Delabie, Cédric Lecluyse, Chesney Buyle, Gilles Callebaut, Liesbet Van der Perre, Lieven De Strycker

**Affiliations:** ESAT-DRAMCO, Ghent Technology Campus, KU Leuven, 9000 Ghent, Belgium; jarne.vanmulders@kuleuven.be (J.V.M.); daan.delabie@kuleuven.be (D.D.); cedric.lecluyse@kuleuven.be (C.L.); chesney.buyle@kuleuven.be (C.B.); gilles.callebaut@kuleuven.be (G.C.); lieven.destrycker@kuleuven.be (L.D.S.)

**Keywords:** wireless power transmission, inductive power transmission, capacitive transducers, RF signals, optical beams, acoustics, unmanned vehicles, circuits, safety, standards, applications

## Abstract

Wireless power transfer provides a most convenient solution to charge devices remotely and without contacts. R&D has advanced the capabilities, variety, and maturity of solutions greatly in recent years. This survey provides a comprehensive overview of the state of the art on different technological concepts, including electromagnetic coupled and uncoupled systems and acoustic technologies. Solutions to transfer mW to MW of power, over distances ranging from millimeters to kilometers, and exploiting wave concepts from kHz to THz, are covered. It is an attractive charging option for many existing applications and moreover opens new opportunities. Various technologies are proposed to provide wireless power to these devices. The main challenges reside in the efficiency and range of the transfer. We highlight innovation in beamforming and UV-assisted approaches. Of particular interest for designers is the discussion of implementation and operational aspects, standards, and safety relating to regulations. A high-level catalog of potential applications maps these to adequate technological options for wireless power transfer.

## 1. Introduction

Wireless power transfer is attractive because of the number of benefits it can bring. Evidently, the fact that connectors can be avoided is convenient when charging devices. Furthermore, a contactless solution can be more reliable, avoiding corrosion and intrusion of dust and moisture. It can provide a hygienic solution as, e.g., medical appliances can be fully sealed and easily disinfected. The autonomy of devices can be prolonged when they can get charged in situ from a shorter (through materials) or longer distance. Several technologies and standards for wireless charging have been developed. The latter has opened the opportunity to reuse chargers for multiple devices. Meanwhile, R&D teams develop new systems and circuits in search of solutions that can increase charging range and efficiency, provide higher power safely, and enable new use cases.

This survey paper provides an in-depth overview and a wide study of wireless power transfer (WPT) systems, based on different technological concepts, and circuits. Further, this paper discusses several techniques to enable efficient and safe operation, standards, and applicable regulations, implementation challenges, commercial systems, and use cases. Some papers have focused on specific applications, e.g., biomedical implants [1]. This paper further expands previous work and presents novel developments, including UV-assisted wireless power transfer, beamforming, and repeater-enhanced deployments. We summarize our insights into a high-level catalog of applications versus candidate technologies for developers.

Methods to extend coverage and increase efficiency, including beamsteering, the introduction of repeaters, and hybrid systems, are here given particular attention. UV-assisted wireless charging presents an interesting option to disrupt the conventional efficiency–range bottleneck. Through this concept, energy can be delivered in the proximity of the device in a safe way, while the unmanned vehicle (UV) itself could recharge opportunistically on renewable energy.

The status of standards and available commercial solutions is further summarized. We also zoom in on challenges related to implementation and safety. An extensive overview of potential applications and their mapping to candidate technologies is given. This is rather unique in covering from milliwatts to megawatts and short to long distances and provides a helpful reference for the developer community.

We structure the WPT solutions according to technological categories as sketched in Figure 1: *electromagnetic* and *acoustic* wave-based power transfer (acoustic power transfer (APT)). Both coupled and uncoupled systems using electromagnetic principles exist. Coupled systems using magnetic or electric fields, are called inductive power transfer (IPT) and capacitive power transfer (CPT) respectively. Wireless power transmission through RF (radio frequency power transfer (RFPT)) and light (laser power transfer (LPT)) are uncoupled systems. Figure 1 includes the section number for easy navigation in the manuscript.

Wireless energy transfer comes with a number of limitations, which depend on the technology. Table 1 provides a preview of the possibilities of each energy transfer approach. Numeric values are excluded here, as these strongly depend on the specific implementation. Essential information is bundled in this overview table to form a global picture of the discussed technologies and their properties. IPT and CPT systems are the most common coupled WPT systems that can deliver up to MWs of power over distances of several mm to cm. Operating frequencies vary from kHz to MHz, which is feasible as these technologies are working in the near field. High efficiencies can be achieved with an inductive or capacitive link, even with high power delivery systems. The main limitation is their short charging distance. The uncoupled LPT and RFPT systems are elaborated in different ways. Existing implementations may comply with the laser safety classes and the Industrial, Scientific and Medical (ISM) band regulations. Opposite, in specific cases, the regulations may not be considered, and extra safety features are adopted. In Table 1 these two approaches in safety considerations are distinguished from each other with a slash. APT often finds its use in application where device miniaturization is important or electromagnetic (EM)-based WPT is difficult. In implantable medical devices, power levels up to a few hundred mW can be delivered safely, while several kW can be transferred through centimeter-thick closed metal constructions. The different technologies are more extensively discussed in the next sections, providing references to both fundamental and recent works.

This paper is organized as follows. In the next section, the most deployed electromagnetic coupled technologies are discussed. In Section 3, electromagnetic uncoupled technologies, including both radio frequency (RF)- and laser-based solutions, are elaborated. We cover acoustic power transfer technologies in Section 4. Section 5 presents technologies that enable improvements in range, power, and efficiency. Next, Section 6 provides an overview of standards and commercial solutions, and Section 7 discusses safety aspects in a regulatory context. Implementation and operational challenges are explained in Section 8. A survey of use cases mapping these to technological options is provided in Section 9. Section 10 discusses current gaps and future trends. This paper is concluded in Section 11 and gives a survey on technological concepts, systems, and circuits, including a preview of possible efficiency improvements and key operational aspects for various use cases.

## 2. Electromagnetic-Coupled Technologies

### 2.1. Overview of an EM-Coupled System

In inductive and capacitive wireless power systems, the transmitters (TX) and receivers (RX) are placed in close proximity to each other. The coupling factor between TX and RX, depending on several design parameters, largely determines the link efficiency. The coupled WPT technologies rely on a general structure depicted in Figure 2. First, a power supply in combination with a transmitter circuit provides an amplified sine wave that can be controlled in both frequency and amplitude. For this purpose, building blocks such as a pre-regulator, power inverter, and wave generator are required. Secondly, the transmitter antenna is coupled with the receiver antenna by means of electric or magnetic fields. Thirdly, the receiver circuit consists of an AC/DC conversion by means of a rectifier, smoothing capacitors, and optionally a switched-mode power supply (SMPS). The building blocks are briefly introduced below.

#### 2.1.1. Power Supply

WPT systems can not directly use the AC mains to power the transmitter circuit, therefore an initial conversion to DC is required. The AC/DC converter can be implemented in several ways and is thus represented in Figure 2 by the DC power supply. Mains-powered systems typically contain a flyback converter to achieve a safe, stable DC voltage. Alternatively, the transmitter can be powered by a battery.

#### 2.1.2. Transmitter Circuit

*Pre-regulator.* In some implementations, a pre-regulator is introduced between the power supply and the inverter with the purpose of controlling the amplitude of the sine wave that will be fed to the antenna. The pre-regulator is not always required since many systems can operate on a fixed voltage. The pre-regulator can be found in the form of SMPSs, such as step down, step up, or buck-boost converters [2,3]. They have very low losses and, hence, are highly efficient.

*Inverter.* Different types of power inverters can be considered in the design. Power amplifiers such as class-A and class-B are not efficient enough for WPT applications. Alternatively, class-C, class-D, and class-E amplifiers can provide a more efficient solution [4]. Mostly, the half-bridge or full-bridge DC/AC inverter is selected for coupled WPT systems. A driver controls one or multiple transistors to obtain an amplified periodic signal. Different types of metal-oxide-semiconductor field-effect transistors (MOSFETs) are found in WPT systems. Microsemi [5] compare the properties of suitable transistor substrates. Low-frequency inverters are typically constructed with Si field-effect transistors (FETs). High-frequency inverters require wide band gap (WBG) electronics to reduce the input capacitance, meaning that silicon carbide (SiC) and gallium nitride (GaN) FETs are appropriate substrates for these higher frequency inverters.

Figure 3 illustrates a transmitter circuit consisting of a pre-regulator, square-wave generator, gate driver, and half-bridge power inverter. This circuit creates an amplified periodic signal. The power FETs are controlled with an appropriated gate driver integrated circuit (IC). The gate driver input is connected to a square wave generator, e.g., an oscillator for a fixed frequency or a voltage-controlled oscillator (VCO) for variable frequency. The pre-regulator can be seen as a DC/DC converter with adjustable output voltage, connected to the power inverter. Changing the pre-regulator output voltage results in a higher amplitude of the square wave. Both the pre-regulator and the oscillator can be controlled with a microcontroller unit (MCU) to change the amount of power transfer, save energy, or optimize the link efficiency.

#### 2.1.3. Compensation Network and Antenna

The matching circuit is required to get the link efficiency as high as possible. The link efficiency is expressed as the power received at the receiver antenna relative to the power transmitted at the transmit antenna. A matching network consists of inductive and capacitive impedances connected in parallel and series. In the literature, the terms primary and secondary tank describes the combination of compensation network and antenna [6].

#### 2.1.4. Receiver Circuit

*Rectifier and smoothing capacitors.* The AC voltage coming from the antenna and compensation network is converted to DC with a rectifier and smoothing capacitors. $R_{L}$, from Figure 2, represents the DC-load. In the literature, an AC load is often used to optimize the link and represents the equivalent load directly connected to the secondary resonant tank. Knowledge about the rectifier circuit allows calculating the corresponding AC load and, especially in strong coupled systems, to increase the efficiency. In particular, there is a certain optimal load to achieve the maximum link efficiency. Changing parameters on the transmitter side modifies the secondary resonance amplitude and consequently also the AC load. Measuring real-time secondary voltages and currents, and communicating them through a feedback link, allows the transmitter to modify the amplitude and frequency to achieve higher link efficiencies.

The DC and AC load are mostly variable, e.g., when charging batteries. The rectification circuits: half-wave and full-wave (with or without voltage doubling) rectifiers are the most commonly used diode circuits for the AC–DC conversion [6]. The minimization of rectifier losses requires diodes with low forward voltages like, e.g., Schottky diodes. A WPT system is often built for a specific application with a known output current and voltage. The parameters of the diode, such as forward current and peak reverse voltage, can be optimized, thus obtaining the lowest forward voltage and reducing the diode power losses. Alternatively, active rectifiers achieve higher efficiencies as they are built with FETs. The drain-source voltage is lower compared to the diode forward voltage, resulting in lower energy losses. Active rectifiers are based on MOSFETs [7] for low frequency switching or GaN FETs for high frequency switching [8]. A semi-bridgeless active rectifier (S-BAR) can also improve efficiency and is built with a combination of diodes and FETs [9].

*DC/DC regulator.* There are multiple reasons to have a regulator on the receiver side. First, the need for a constant voltage (CV) to power, e.g., an MCU. Secondly, a constant current (CC) to power, e.g., a LED. Thirdly, a combination of CV and CC to charge, e.g., lithium batteries with a specific CC-CV charge profile. Both an low-dropout regulator (LDO) and an SMPS can be used as a DC/DC regulator. Generally, SMPSs are more energy-efficient.

### 2.2. Inductive Coupling

Magnetic links are basically DC/DC converters built around a loosely coupled transformer. A typical system consists of magnetic coupling between a secondary and primary coil driven by an alternating current. The two coils form a coreless transformer, where the coupling is determined by the coupling factor $k_{m}$. An alternating current flowing through the primary coil produces an alternating magnetic field, following Ampére’s law [6]. The varying magnetic flux at the secondary coil induces an electromotive force by Faraday’s law of induction [6], hence the naming IPT in the literature. The most important aspect, to build an efficient IPT system, is explained here, while the building blocks of a typical system have been discussed above. The efficiency is largely determined by the resistive coil losses and the coupling factor between the two coils. The coupling factor can be calculated with Equation (Equation 1) and depends on the mutual inductance *M* (in H) and the self-inductance of the transmitter and receiver coils $L_{1}$ and $L_{2}$ (in H) respectively.
(1)$$k_{m}=\frac{M}{\sqrt{L_{1}\;\cdot \;L_{2}}}$$

The subscript *m* is used to denote that it is working on the principles of alternating magnetic fields.

If the distance between the coils increases, the mutual inductance will quickly decrease, resulting in a lower coupling factor. This significant reduction in coupling with increasing distance indicates that WPT via inductive coupling can only cover small distances. A basic IPT system is depicted in Figure 4 and contains the AC source, the two coils $L_{1}$-$L_{2}$, and the AC load $R_{L}$. This equivalent AC load represents the rectifier, DC/DC regulator, and DC load.

The link efficiency is key and should be maximized at all times. It describes the relationship between the power from the secondary coil relative to the power from the primary coil. In order to increase the efficiency, the AC load needs to be known, especially in strongly coupled systems. In particular, there is a certain optimal load to achieve the maximum link efficiency. The load factor *a* and the quality factors from the primary and secondary coil, respectively, $Q_{1}$ and $Q_{2}$ are given in (Equation 2). The optimal load factor $a_{max}$ depends only on the coil quality factors and coupling factor, given in (Equation 3) [6].
(2)$$Q_{1}=\frac{\omega L_{1}}{R_{L_{1}}},\;Q_{2}=\frac{\omega L_{2}}{R_{L_{2}}}\;\mathrm{and}\;a=\frac{\omega L_{2}}{R_{L}}$$
(3)$$a_{max}=\frac{Q_{2}}{\sqrt{1+Q_{2}^{2}+k^{2}\;\cdot \;Q_{1}\;\cdot \;Q_{2}}}$$

The optimal load $R_{L,optimal}$ can then be found based on (Equation 3) and (Equation 2). Assuming this optimal load is connected to the secondary coil, the link efficiency is entirely determined by the coupling factor in combination with the equivalent coil resistances, $R_{L_{1}}$ and $R_{L_{2}}$, the ideal self-inductance values, $L_{1}$ and $L_{2}$, and the angular frequency ω. The maximum link efficiency for a non-resonant and resonant secondary tank, where $X=k^{2}\;\cdot \;Q_{1}\;\cdot \;Q_{2}$, are given in (Equation 4) and (Equation 5), respectively.
(4)$$\eta_{link,non-res,max}=\frac{X}{2+X+2\sqrt{1+Q_{2}^{2}+X}}$$
(5)$$\eta_{link,res,max}=\frac{X}{{\left(1+\sqrt{1+X}\right)}^{2}}$$

For resonant magnetic WPT, an additional capacitor $C_{2}$ is required on the secondary side to make it resonant and to achieve the maximum link efficiency given by (Equation 5). Poor link efficiency in the case of low coupling is due to the secondary leak inductance $L_{2}\left(1-k^{2}\right)$. The reason is that the impedance from the leak inductance increases when the coupling factor decreases. Consequently, less energy will be available on the load $R_{L}$. Therefore, this leakage inductance is canceled with a capacitor $C_{2}$ as shown in Figure 5.

Figure 6 shows the maximum achievable link efficiency. The efficiency of the non-resonant receiver in Equation (Equation 4) depends on the secondary quality factor which results in a lower achievable maximum efficiency. It can be noted that for the non-resonant system, the efficiency decreases with increasing $Q_{2}$. In general, lower coupling factors or lower coil quality factors lead to lower overall power transfer efficiencies. Since the quality factor of a coil depends on the frequency, the link efficiency also depends on the operating frequency.

During the wireless transfer, the power transmitter can adjust the frequency or the amplitude depending on whether more or less power is required [2]. For example, when a battery is almost entirely charged and the condition for $a_{max}$ is no longer met, the $R_{L}$ or ω can be adjusted by changing the frequency or primary voltage. This should result again in an efficient link and this technique can also be used to prevent overvoltage at the receiver side. The transmitter has no view of the receiver voltage and output power. It is therefore required to feed these values back to the transmitter. This is mostly realized with capacitive or resistive load modulation. Capacitive load modulation changes the resonance frequency of the secondary tank by switching an additional capacitor. Resistive load modulation switches a resistor in parallel with the secondary tank. Both approaches result in an amplitude-shift keying (ASK) signal that is demodulated at the transmitter side. The transmitter reads the signal and is able to adjust the transmit power by changing the frequency or amplitude. In the literature, this is known as on-line adjustment of the driver output power [6].

It is feasible to make an inductively coupled system based on the aforementioned design principles that transfers power up to several tens of watts. Standards however provide a range of additional, yet sometimes indispensable features such as interoperability, safety measures, communication between charger and charged device, etc. The Qi specification [2], for example, ensures safe energy transfer through thermal shutdown protection, foreign object detection, and over-voltage AC clamp protection. It is strongly recommended that the developer installs such safety features as well as a dedicated communication system.

### 2.3. Magnetic Resonance Coupling

Magnetic resonance coupling (MRC) works on the same principle as IPT: two coils with series or parallel capacitors on both transmitter and receiver sides, form a coupled WPT system. Figure 7 illustrates the link structure, where $k_{mrc}$ indicates the coupling factor between transmitter and receiver coil.

The main difference between inductive coupling and magnetic resonance coupling is the lower coupling factor in magnetic resonance coupling (MRC) systems. An IPT implementation uses tightly coupled coils with typical coupling factors $k_{m}$ > 0.3 [2], unlike MRC systems with $k_{mrc}$ < 0.1 [10]. Such systems are efficient, even with low coupling factors. The alignment requirements are less stringent, thus giving more spatial freedom to position the receiver relative to the transmitter. To optimize energy transfer with low coupling, the classical efficiency optimization described in Section 2.2 could be used. However, the critical coupling method is preferably used, where both receiver and transmitter are tuned to the same resonance frequency. Different studies describe the efficiency of an MRC system [10,11,12]. This overview elaborates on the transfer coefficient, link efficiency, and overall efficiency.

Figure 8 illustrates an extension of Figure 7 with the transmitter and receiver coil replaced by the equivalent T-model. The series resistances from coils $L_{1}$ and $L_{2}$ are presented by $R_{L_{1}}$ and $R_{L_{2}}$, respectively. The internal series resistor from the power supply $V_{S}$ is represented by $R_{S}$.

We assume both transmitter and receiver are tuned to the same resonance frequency $\omega_{0}$, with
(6)$$\omega_{0}=\frac{1}{\sqrt{L_{1}C_{1}}}=\frac{1}{\sqrt{L_{2}C_{2}}}.$$

By applying the Kirchhoff’s voltage law (KVL) in Figure 8, the expressions (Equation 7) and (Equation 8) are derived.
(7)$$V_{S}=\left(R_{S}+R_{L_{1}}+j\omega L_{1}+{\left(j\omega C_{1}\right)}^{-1}\right)I_{T}+j\omega MI_{R}$$
(8)$$0=j\omega MI_{T}+\left(j\omega L_{2}+{\left(j\omega C_{2}\right)}^{-1}+R_{L_{2}}+R_{L}\right)I_{R}$$

The transmitter and receiver series resistances, $R_{T}$ and $R_{R}$ respectively, are given by (Equation 9).
(9)$$R_{T}=R_{S}+R_{L_{1}},\;R_{R}=R_{L_{2}}+R_{L}$$

The mutual inductance *M* is equal to $k_{mrc}\sqrt{L_{1}L_{2}}$. The transmitter and receiver quality factor $Q_{T}$ and $Q_{R}$ are given in (Equation 10).
(10)$$Q_{T}=\frac{\omega L_{1}}{R_{T}},\;Q_{R}=\frac{\omega L_{2}}{R_{R}}$$

Transforming (Equation 7) and (Equation 8) gives the transmitter and receiver current in (Equation 11).
(11)$$I_{T}=\frac{1}{1+k_{mrc}^{2}Q_{T}Q_{R}}\frac{V_{S}}{R_{T}},\;I_{R}=\frac{-j}{\sqrt{R_{T}R_{R}}}\frac{k_{mrc}\sqrt{Q_{T}Q_{R}}}{1+k_{mrc}^{2}Q_{T}Q_{R}}V_{S}$$

The transfer coefficient Π, given in (Equation 12), shows how the distance and coils affect the secondary voltage.
(12)$$\Pi =\frac{k_{mrc}\sqrt{Q_{T}Q_{R}}}{1+k_{mrc}^{2}Q_{T}Q_{R}}$$

$I_{R}$, and thus the transferred power, is maximized when (Equation 13) is fulfilled. A coupling coefficient smaller, equal or greater than $1/\sqrt{Q_{T}Q_{R}}$ is called undercoupled, critically coupled, or overcoupled, respectively [10].
(13)$$k_{mrc}=\frac{1}{\sqrt{Q_{T}Q_{R}}}$$

The efficiency $\eta_{link}$ is determined by the total received power versus the overall consumed power. The power consumed at the transmitter and receiver side is respectively $P_{T}=I_{T}^{2}\;\cdot \;R_{T}$ and $P_{R}=I_{R}^{2}\;\cdot \;R_{R}$ and gives in combination with (Equation 11) the link efficiency in (Equation 14).
(14)$$\eta_{link}=\frac{P_{R}}{P_{T}+P_{R}}=\frac{k_{mrc}^{2}Q_{T}Q_{R}}{1+k_{mrc}^{2}Q_{T}Q_{R}}$$

Similarly, the overall efficiency describes the consumed power in the load relative to the transmitted power (Equation 15).
(15)$$\eta_{overall}=\frac{P_{L}}{P_{T}+P_{R}}=\frac{R_{L}}{R_{R}}\frac{k_{mrc}^{2}Q_{T}Q_{R}}{1+k_{mrc}^{2}Q_{T}Q_{R}}$$

The power to the load $P_{L}=I_{R}^{2}\;\cdot \;R_{L}$ can be rewritten according to the transfer coefficient Π in (Equation 16). $P_{L}$ is proportional to the square of the transfer coefficient.
(16)$$P_{L}=\frac{R_{L}}{R_{T}R_{R}}\frac{k_{mrc}^{2}Q_{T}Q_{R}}{{\left(1+k_{mrc}^{2}Q_{T}Q_{R}\right)}^{2}}V_{S}=\frac{R_{L}}{R_{T}R_{R}}\Pi^{2}V_{S}$$

Figure 9 depicts the transfer coefficient and efficiency as a function of the coupling and quality factors. If $k_{mrc}{\left(Q_{T}Q_{R}\right)}^{1/2}=1$, the highest transfer coefficient is achieved. Higher values $k_{mrc}{\left(Q_{T}Q_{R}\right)}^{1/2}>1$ result in higher efficiencies. The transfer coefficient Π, which is proportional to the receiver current, indicates that the too low or too high values for $k_{mrc}{\left(Q_{T}Q_{R}\right)}^{1/2}$ reduce the secondary current. In most practical systems, $k_{mrc}$ is low (around $10^{-2}$) and quality factors around $10^{2}$ are required. Further, *Q* impacts the dissipation of energy in the resonant circuit. High values give low dissipation although low *Q* values make the circuit broadband [11]. Since MRC systems typically work on one fixed predefined frequency (mostly 6.78 MHz) high-quality factors are recommended. To regulate the power at the receiver only the amplitude $V_{S}$ can be adjusted during operation.

The benefit of an MRC-based implementation is the opportunity to provide energy to multiple receivers. The higher spatial freedom and operating area make this technology suitable for applications where alignment is difficult or impossible. The main drawback is the increased design complexity. Both the use of GaN FETs and zero voltage switching (ZVS) class-D amplifiers are recommended to achieve a high efficiency.

### 2.4. Electrodynamic Coupling

An electrodynamically coupled WPT system delivers energy from a transmitter coil to an electromechanical receiver using low-amplitude and low-frequency magnetic fields and is depicted in Figure 10. The energy transfer distance is typically a few millimeters to centimeters. The transmitter induces a torque on the permanent magnet situated at the receiver side. The amount of torque depends on the distance and orientation between the transmitter and receiver. The mechanical energy is converted into electrical energy using transducers such as electrodynamic, piezoelectric, or electrostatic transducers. The electromechanical receiver is modeled as a damped spring-mass system with *k* being the spring constant, $m_{\mathrm{eff}}$ the mass, $b_{m}$ the damping coefficient which represents mechanical losses, and $b_{e}$ the electrical power delivered to the load $R_{L}$. In [13] a model for the system efficiency is proposed.

The amount of transferred energy is typically only a few hundred microwatts. Compared to the inductively coupled WPT systems, it has a very low efficiency in the range of 4.1% to 12% [13]. Due to the low operating frequency, coils with a huge number of windings are required. On the other hand, this technology relies on the mechanical resonance of the electromechanical receiver for energy transmission, which enables the use of much lower resonance frequencies (e.g., 40 kHz) than other WPT technologies. The strength of the magnetic field can therefore be higher without exceeding the safety limits for human exposure [13,14].

### 2.5. Capacitive Coupling

This approach is based on electric fields between plates to transfer energy. A typical system consists of two capacitors. Four metal plates form the left and right side of capacitors $C_{A}$ and $C_{B}$, as shown in Figure 11. Supplying this system with an alternating voltage generates an electric field. At the receiver side, the electric field induces a current. When operating at high enough frequencies, these capacitors will act as if they were conductors. The advantages of this technology are the lower cost of the transmitter and receiver, the reduced influence on efficiency in the presence of surrounding metal objects, and reduced size compared to IPT. The distance between the two plates varies from a few millimeters to tens of cm. Due to the maximal field strength, in the form of electric breakdown, the gap power density of CPT in air is 400 times lower than IPT. WPT over longer distances, e.g., 15 cm, requires several unfavorable techniques, such as high voltages, large plates, high switching frequencies, and high electric fields. The latter causing safety concerns for the surrounding area [15,16].

The major challenge for CPT is the low coupling. This has been partially solved by using compensation circuits, as will be explained later on. However, as these types of systems are usually designed for very specific requirements or applications, it is difficult to use these for similar systems with different loads without impacting performance. An approach to cope with variable loads or changing distances is proposed in [17]. Nevertheless, more research is needed in this area.

The link efficiency is here derived. The capacitive coupler can be represented by several equivalent schemes, including the Pi-model (Figure 12a) and the equivalent model with voltage controlled current sources (VCCS) (Figure 12b).

To achieve energy transfer via capacitive coupling, at least two metal plates are needed to transmit and capture the electric fields. The plate structure of the CPT system affects the capacity of the plates and consequently the capacitive coupling coefficient $k_{e}$, which can be calculated by (Equation 17). Note, the use of the subscript *e* to denote that it is operating on electric fields. $C_{in_{1}}$, $C_{in_{2}}$, and $C_{M}$ are the resulting primary, secondary, and mutual capacitance of the equivalent Pi-model representation of the CPT model [15,18,19].
(17)$$k_{e}=\frac{C_{M}}{\sqrt{C_{1}\;\cdot \;C_{2}}}$$

The scheme from Figure 11 is in practice extended with compensation networks between the AC source and the primary plates and between the secondary plates and the load. The primary and secondary compensation networks are represented by respectively $C_{ext_{1}}$, $L_{1}$ and $C_{ext_{2}}$, $L_{2}$. An equivalent capacitive coupler is made by $C_{1}$ and $C_{2}$. They form the combination of the coupling capacitors ($C_{int_{1}}$ and $C_{int_{2}}$) and the capacitors used in the compensation circuit ($C_{ext_{1}}$ and $C_{ext_{2}}$). Figure 11, combined with the compensation networks and the VCCS equivalent model, give the equivalent scheme in Figure 13.
(18)$$\begin{array}{c} C_{1}=C_{in_{1}}+C_{ex_{1}},\;C_{2}=C_{in_{2}}+C_{ex_{2}} \end{array}$$

The equivalent resistances of compensation coils ($R_{L1}$, $R_{L2}$) and capacitances ($R_{C1}$, $R_{C2}$) combine to $R_{L_{1}}$ and $R_{L_{2}}$.
(19)$$\begin{array}{c} R_{L_{1}}=R_{L_{1}}//R_{C_{1}},\;R_{L_{2}}=R_{L_{2}}//R_{C_{2}} \end{array}$$

The link efficiency $\eta_{link}$ is further derived using the equivalent scheme from Figure 13 and consists of the power consumed by the load with respect to the overall consumed power.
(20)$$\eta_{link}=\frac{\frac{|V_{C2}{|}^{2}}{R_{L}}}{\frac{|V_{C1}{|}^{2}}{R_{L_{1}}}+\frac{|V_{C2}{|}^{2}}{R_{L_{2}}}+\frac{|V_{C2}{|}^{2}}{R_{L}}}$$

The secondary current can be written as presented in (Equation 21).
(21)$$I_{2}=\frac{V_{C_{2}}}{R_{C_{2}}//R_{L}}=j\omega C_{M}V_{C_{1}}$$

Substituting (Equation 21) into (Equation 20) gives (Equation 22).
(22)$$\eta_{link}=\frac{1}{\frac{1}{{\left(R_{C_{2}}//R_{L}\right)}^{2}\omega^{2}C_{M}^{2}}\frac{R_{L}}{R_{L_{1}}}+\frac{R_{L}}{R_{L_{1}}}+1}$$

The quality factors for $C_{1}$ and $C_{2}$ are shown in (Equation 23), as also the load factor *a*.
(23)$$Q_{1}=\omega C_{1}R_{L_{1}},\;Q_{2}=\omega C_{2}R_{L_{2}}\mathrm{and}a=\frac{R_{L_{2}}}{R_{L}}$$

After substituting (Equation 23) and (Equation 17) in (Equation 22), the link efficiency is given in (Equation 24).
(24)$$\eta_{link}=\frac{1}{\frac{a+\frac{1}{a}+2}{k_{e}Q_{1}Q_{2}}+\frac{1}{a}+1}$$

As IPT, a CPT system has an optimal load factor $a_{max}$
(25)$$a_{max}=\sqrt{1+k_{e}^{2}Q_{1}Q_{2}}$$

The maximum link efficiency for an CPT system is consequently given by (Equation 26).
(26)$$\eta_{link,max}=\frac{k_{e}^{2}Q_{1}Q_{2}}{{\left(1+\sqrt{1+k_{e}^{2}Q_{1}Q_{2}}\right)}^{2}}$$

The same maximum link efficiency equation as for an IPT system (Equation 5) can be obtained by changing $k_{e}^{2}Q_{1}Q_{2}$ by the variable *X*. Consequently, Figure 6 again shows how the maximum link efficiency depends on the quality factors $Q_{1}$, $Q_{2}$, and the coupling factor $k_{e}$.

In the following subsections, typical plate structures, circuit topologies, and compensation circuits are introduced.

#### 2.5.1. Plate Structures

The literature describes four structures.

*Two-Plate Structure.* A two-plate plate structure is similar to the composition of a capacitor. One plate is connected to the transmitter and the other plate to the receiver. In a setup as shown in Figure 14a, energy is transferred via the two plates $P_{1}$ and $P_{2}$ and the returning path of current is via a conductive path, e.g., the Earth. In railway applications, this is possible via the wheels connected to the track. In this way, CPT can be a replacement for the maintenance-intensive pantograph [20].

*Four-Plate Parallel Structure.* This is the most common CPT structure, consisting of two pairs of two parallel plates that form the main capacitive coupling. In practice, there are six capacitors that form the coupling: two main capacitors ($C_{13}$, $C_{24}$), two leakage capacitors ($C_{12}$, $C_{34}$), and two cross-coupling capacitors ($C_{23}$, $C_{14}$) (see Figure 14b). When the plates are well-aligned and close to each other, the main capacitor dominates the total capacitance. If these requirements are not fulfilled, the influence of the parasitic capacitors will be larger, resulting in a lower coupling quality.

*Four-Plate Stacked Structure.* A stacked four-plate system, shown in Figure 14c, is more compact than the parallel structure and is more resilient to rotary misalignment. The outer plates $P_{1}$ and $P_{3}$ of the structure are larger than the inner plates $P_{2}$ and $P_{4}$ in order to establish capacitive coupling between the outer plates. However, this reduces the main capacitance and increases the parasitic capacitance, as the coupling area of the outer plates is smaller. This results in a lower coupling quality compared to the parallel structure [19,21,22].

*Six-Plate Structure.* This structure is a four-plate parallel structure with two extra shielding plates to provide better protection against electric fields, as seen in Figure 14d. The additional plates do not take part in the power transfer and are only coupled via parasitic capacitance. Studies have shown that this approach improves safety and efficiency [23,24,25,26].

#### 2.5.2. Circuit Topologies

As mentioned above in Section 2, the full-bridge inverter is the most common for WPT systems, which also applies to CPT. The reason is the low complexity of implementation and parameter design, as well as its robustness [27]. Furthermore, high input voltages can be applied without the appearance of major voltage stress across the switching components compared to other topologies [28]. Two other primary circuit topologies are addressed in the literature, the PWM and power amplified-based converters.

*PWM-based converters*, such as buck-boost, Cuk, single-ended primary-inductor converter (SEPIC), ZETA, and push-pull, are single switch converters. The LC storage tank can be adapted so that the capacitor creates a primary and secondary side in the converter. In this way, CPT is possible. Figure 15b shows such a modified sepic converter [29,30]. The main advantage of this approach is the reduced sensitivity against varying coupling caused by misalignment or varying distances between plates. However, the distance over which power can be transferred is limited due to the use of one active switch. The specifications of this switch determine the maximum power transfer. Another disadvantage of PWM-based converters is that soft-switching is not possible in every load condition, which increases the electromagnetic interference electromagnetic interference (EMI) and decreases the efficiency [29].

*Power amplifier-based converters* are used in a variety of applications, from audio systems to motor drives. By using a resonant LC tank, the converter can operate in zero voltage switching mode [31]. A modification in the LC tank of a power amplifier-based converter, such as class-D, class-E, class-EF, or ϕ, ensures that they can realize CPT [32,33]. Figure 15c shows a class-E converter with the LC tank modified to transfer energy. To obtain zvs mode, accurate tuning of the resonant LC tank is required. The capacitor of the LC tank is formed by the capacitive coupling between the primary and secondary sides. This makes the system very sensitive to changes in system parameters such as distance between the plates, which can result in loss of ZVS. Power amplifier-based converters are therefore best used in static applications.

#### 2.5.3. Compensation Circuits

To achieve maximum power transfer, the system needs to operate at the resonant frequency. Since the resulting capacitance between the plates is very small and consequently high frequencies are required, compensation circuits are used to lower this resonant frequency. However, this does not solve the entire problem, since these types of systems are always very specifically designed for certain requirements or applications. This will make it difficult to apply the same system to different loads, as it will affect the performance of the system. A solution to problems such as variable loads or changing distances is proposed in [17]. Nevertheless, more research is needed in this area.

*L compensation.* An inductor is added to the system, either on the primary or on the secondary side. Where it is added and whether it is in series or parallel with coupling plates determines the properties of the system [15,34,35,36]. This is the most basic compensation circuit and provides a maximum gain of 1 at the resonant frequency, resulting in maximum power transfer [34]. An example of a primary series resonance compensation circuit can be found in Figure 16a.

*LC compensation.* Adding just a coil does not solve the problem of low coupling. Therefore, a capacitor is often added in parallel across the mutual capacitance on both primary and secondary sides. This increases the resulting capacitance and makes the system less sensitive to varying distances between transmitter and receiver. An example is a double-sided LC compensation network, shown in Figure 16b [15,34,36].

*Multistage L-sections.* By adding more coils and capacitors, multistage networks can be created. These networks can respond more precisely to the objectives of the system, such as achieving greater powers over greater distances. Often, these networks create a voltage gain on the primary side and a current gain on the secondary side in order to maximize power transfer. The voltage or current gain depends on the structure of the L-section, as shown in Figure 16c [15,34,36].

## 3. Electromagnetic Uncoupled Technologies

This section provides an overview of uncoupled WPT technologies. More specifically, RF and laser power transmission are discussed. Unlike coupled WPT, the transmitters and receivers are placed in the far field. The link budget and typical losses are examined for both options. These systems are also referred to as radiative WPT, as opposed to the non-radiative coupled systems discussed in previous sections.

### 3.1. RF Power Transfer

RF power transmission, sketched in Figure 17, has recently received significant interest because of the increased use of wireless, low-power devices. Radio waves offer the advantage of being ubiquitous and hence energy can be obtained over a large area, even in inaccessible places. However, RF energy has the lowest power density when compared to other ambient energy harvesting technologies such as solar, thermal, and vibration energy [37]. The power density of RF energy varies between 0.2 nW/cm${}^{2}$ and 1 μW/cm${}^{2}$ [38]. Practically, this results in power in the range of μW at the receive side.

RF power can be harvested from ambient or dedicated sources. Ambient RF sources refer to transmitters that are not intentionally made for energy transfer, such as TV towers, Wi-Fi, and cellular communication systems [39]. The energy harvested from these sources can be seen as free as the transmitters serve another purpose and the radio signals that are not received would otherwise remain unused. The key challenge lies in the extremely low power density that is available to the energy harvesting nodes, as discussed in research surveys [40,41,42]. At a large scale, the frequency bands with the highest power density may vary, which makes it impossible to design a single, optimal energy harvesting circuit. As opposed to single-band rectennas, multiple frequency bands can be exploited to maximize harvested power. The particular case of simultaneous wireless information and power transmission (SWIPT) has received a lot of attention recently. We therefore refer the reader to focused SWIPT studies, e.g., [43,44], and in this paper consider wireless energy transfer as a desired standalone feature. Dedicated RF sources, on the other hand, can be deployed to increase power transfer to, e.g., sensor and actuator nodes. Although higher power levels can be achieved through dedicated sources, in practice the levels are often limited by regulations and safety levels. Consequently, the main advantages are that power can be pointed to the receiver increasing the overall efficiency and that a more predictable energy supply is available at the nodes. Again, multiple frequency bands can be exploited to further increase the harvested power, which has shown to be critically important in some applications to meet regulatory constraints [45].

The efficiency of RF power transfer is determined from the path loss model shown in Figure 18. The transmit power $P_{t}$ is sent to the antenna and converted to electromagnetic transmission power $P_{wt}$. Losses such as matching, conduction, and dielectric losses are taken into account in the transmit efficiency $\eta_{t}$. The electromagnetic signals propagating to the receiver are affected by path loss, antenna properties such as the gain, directivity, polarization of both the transmit and receive antenna, and the wireless environment. $P_{wr}/P_{wt}$ determines the efficiency of the wireless link $\eta_{w}$. A power reduction also occurs at the receiving antenna due to matching, conduction, and dielectric losses, indicated by $\eta_{r}$. The power entering the receiver is rectified, converted with a boost converter, and then stored in a buffer or used for a load, depending on receiver type. The efficiencies of the power conversions $P_{r}$ to $P_{rec}$, $P_{rec}$ to $P_{out}$, and $P_{psu}$ to $P_{t}$ are described with, respectively $\eta_{rec}$, $\eta_{reg}$ and $\eta_{pa}$. To quantify these, the electronic components at the transmitter and receiver must be known.

It is important to take into account differences between the near-field and the far field, described as the Fresnel region and the Fraunhofer region respectively, and shown in Figure 19. The limits are described as a function of the distance between transmitter and receiver *R* (in m) and the wavelength of the signal λ (in m). The Fresnel region is located between distances $R_{1}$ and $R_{2}$, while the Fraunhofer region starts from distance $R_{2}$. The reactive near-field region is not discussed, as it is so close to the antenna that WPT is better performed with the coupled systems discussed in Section 2. The following formulas can be used [38]:(27)$$R_{1}=0.62\sqrt{\frac{D^{3}}{\lambda }}\mathrm{and}R_{2}=\frac{2D^{2}}{\lambda },$$
where *D* is equal to the largest dimension of the antenna. For this WPT technique, the applications are usually in the far field, as the distance to the near-field is usually very small. As a result, we limit the theoretical elaboration to the far field.

Assume $P_{t}$ the transmit power before the matching network of the transmitter and $P_{r}$ the received power after the matching network of the receiver. The link budget calculation between them is based on an extensive Friis transmission formula from [46] and is only valid in the far field. Furthermore, only a line of sight (LoS) component is taken into account, and therefore no reflected components are included, i.e., free space. Figure 20 illustrates the various parameters used in the model, which are [46]: (i) $\eta_{cdt}$ and $\eta_{cdr}$: radiation efficiency of the transmitting and receiving antenna, containing the conduction $\eta_{c}$ and dielectric $\eta_{d}$ efficiencies, (ii) $\Gamma_{t}$ and $\Gamma_{r}$: voltage reflection coefficient at the input terminals of the antenna, (iii) λ: wavelength of the RF signal, (iv) *R*: distance between TX and RX, (v) $D_{t}\left(\theta_{t},\phi_{t}\right)$ and $D_{r}\left(\theta_{t},\phi_{t}\right)$: directivity in function of the azimuth and elevation of the TX and RX antenna, and (vi) ${\hat{\rho }}_{t}$ and ${\hat{\rho }}_{r}$: polarization unit vector of the TX and RX.

The extensive Friss transmission formula is given by [46]:(28)$$\begin{array}{c} \frac{P_{r}}{P_{t}}=\left(1-|\Gamma_{t}{|}^{2}\right)\left(1-|\Gamma_{r}{|}^{2}\right){\left(\frac{\lambda }{4\pi R}\right)}^{2}\;G_{t}\left(\theta_{t},\phi_{t}\right)G_{r}\left(\theta_{t},\phi_{t}\right){\left|{\hat{\rho }}_{t}\;\cdot \;{\hat{\rho }}_{r}\right|}^{2} \end{array}$$

If the load impedance is not equal to the corresponding signal source impedance, matching losses occur. This is taken into account in both transmitter and receiver using the voltage reflection coefficient Γ and can be calculated using the antenna input impedance $Z_{A}$ and the impedance of the coupled electronics $Z_{L}$ (transmission lines, receiver circuits, transmitter circuits). These values are given in Ω and depend on the used antennas and components.
(29)$$\Gamma =\frac{Z_{L}-Z_{A}}{Z_{L}+Z_{A}}$$

The path loss PL occurring when an electromagnetic signal propagates also affects the link budget. This loss depends on the distance *R* and the wavelength of this signal λ.
(30)$$PL={\left(\frac{\lambda }{4\pi R}\right)}^{2}$$

The combination of directivity between transmitter and receiver can also cause impact the received power. This directivity $D\left(\theta ,\phi \right)$ is antenna dependent and can be defined as a function of spherical coordinates. With the combination of the radiation efficiency $\eta_{cd}$ and the directivity, the gain $G\left(\theta ,\phi \right)$ can be determined:(31)$$G\left(\theta ,\phi \right)=\eta_{cd}\;\cdot \;D\left(\theta ,\phi \right)$$

This translates into a total gain efficiency $\eta_{gain}$.
(32)$$\eta_{gain}=G_{t}\left(\theta_{t},\phi_{t}\right)G_{r}\left(\theta_{t},\phi_{t}\right)$$

The polarization of both antennas is also key and can lead to losses. The received power can be reduced to zero when there is no match at all. The polarization of an antenna can be described using the polarization unit vector $\hat{\rho }$, which is a vector perpendicular to the direction of propagation for both transmitter and receiver. There are four typical polarizations: random, linear, circular, and elliptical, which can be combined at transmitter and receiver. The polarization loss factor PLF is described by the inner product of these unit vectors. An example is shown in Figure 21.
(33)$$PLF={\left|{\hat{\rho }}_{t}\;\cdot \;{\hat{\rho }}_{r}\right|}^{2}$$

The polarization unit vector of left-hand circular polarized (LHCP) and right-hand circular polarized (RHCP) antennas can be described as:(34)$${\hat{\rho }}_{LHCP}=\frac{{\hat{a}}_{x}+j\;{\hat{a}}_{y}}{\sqrt{2}},\;{\hat{\rho }}_{RHCP}=\frac{{\hat{a}}_{x}-j\;{\hat{a}}_{y}}{\sqrt{2}}.$$

To calculate the total link budget, the power amplifier, rectifier, and boost converter losses must also be taken into account. These losses depend on the electronic design and can often be deduced from datasheets. The following formula is obtained for the overall efficiency from the power supply $P_{psu}$ to the received power $P_{out}$:(35)$$\frac{P_{out}}{P_{psu}}=\frac{P_{t}}{P_{psu}}\;\cdot \;\frac{P_{r}}{P_{t}}\;\cdot \;\frac{P_{rec}}{P_{r}}\;\cdot \;\frac{P_{out}}{P_{rec}}=\eta_{pa}\;\frac{P_{r}}{P_{t}}\;\eta_{rec}\;\eta_{reg}$$

The power delivered by the RF power transfer system hence highly depends on the distance between the receiver and the RF source(s), the source’s transmit power, frequency, antenna gains, and overall conversion efficiency of the energy harvester. For example, the effective isotropic radiated power (EIRP) of a 2.4 GHz Wi-Fi access point in Europe is limited to 100 mW (or 20 m). For a distance of 10 m, receiver antenna gain of 2.15 dBi, and overall energy conversion efficiency of 40% at the energy harvester, the received power in a free-space environment would be 64 nW.

### 3.2. Laser Power Transfer (LPT)

Similar to RF, dedicated systems for optical power transfer can be set up, called optical wireless power transmission (OWPT). The indoor example can be seen as an OWPT system, where it is clear that the received power with a small photovoltaic (PV) is too low due to the diverse and diffusive character of the light source. Replacing the light source with a laser opens an opportunity since the light of a laser is contained in a narrow beam with a high power density. Consequently, the laser beam can be received using a small surface at distances far away [47]. This setup is called LPT and can provide a few mWs to even several kWs of power to a device. Typically, high-intensity laser power beam (HILPB) systems are used, since these LPT techniques are mainly applied for relatively high power applications (up to several kilowatts). This leads, in addition, to the higher efficiency of the PV element [48].

As illustrated in Figure 22, an HILPB transmitter system consists of a laser emitting a monochromatic beam of light, followed by beamshaping with a set of optics, after which the beam is directed to the remote PV array. The beam director is an important element since the alignment in an LPT setup determines its actual transfer. Especially with portable devices, an additional complexity is created as object localization is required, which is very similar to dedicated RF systems. At the receiver, the PV array converts the laser light back into electricity, based on the photoelectric effect. It is important for efficiency to have a good match in terms of used wavelength and beam intensity between the laser and PV element [49].

This technology focuses on far-field power transmission in the range of several meters to several tens of kilometers. The biggest drawbacks are the need for an los link and its low efficiency, whereby the overall system efficiency is currently around 10–20%. A simplified model is proposed to determine the overall efficiency of this system, which is a combination of the laser efficiency, including the power supply and the electrical-to-optical conversion, the atmospheric losses due to scattering and absorption, the PV array with its optical-to-electrical conversion efficiency, and the power controller, as illustrated in Figure 23.

The voltage conversion can take place with a dedicated circuit to provide the appropriate DC voltage to the laser. This results in a typical power supply with efficiencies $\eta_{pc_{1}}$ around 80–90%.
(36)$$\eta_{pc_{1}}=\frac{P_{driver}}{P_{psu}}$$

Heat generation takes place during the electrical-to-optical conversion by the supplied laser, causing a drastically lower generated optical power $P_{laser,opt}$ in comparison with the consumed power of the laser $P_{driver}$. Since the optical power of a laser is often specified by the manufacturer, the laser efficiency can be obtained via (Equation 37) [50].
(37)$$\eta_{laser}=\frac{P_{laser,opt}}{P_{driver}}$$

The optical output power is determined using (Equation 38) [51].
(38)$$P_{laser,opt}=\zeta \frac{hv}{q}\left(I-I_{th}\right)$$

The supplied current *I* must be greater than the threshold current of the laser $I_{th}$. ζ is the modified coefficient and indicates the fraction of the generated photons that contribute to the output beam. *h* is Planck’s constant and *q* is the electronic charge constant. The laser frequency is represented by *v*.

The propagation is affected by attenuation and scattering follows, which in turn causes power reduction of the laser beam. The attenuation is a function of the distance, wavelength, and visibility, impacted by the type and density of the medium (e.g., fog, rain, smoke, etc.) in which the photons travel. When using lasers (monochromatic light) in a homogeneously distributed medium where no influence from the incident flux on the atoms or molecules of that medium is observed and when reflections are not taken into account, the attenuation can be described using the Beer–Lambert law [52,53]. For gases the link efficiency $\eta_{link}$ is equal to:(39)$$\eta_{link,gas}=\frac{P_{rec,opt}}{P_{laser,opt}}=e^{-\beta_{e}\left(\lambda \right)d},$$
with $\beta_{e}\left(\lambda \right)$ being the extinction coefficient of a particular gas at a certain wavelength λ and *d* the transmit distance. The extinction coefficient characterizes how easily a medium can be penetrated by a light beam and depends on both absorption $\beta_{a}\left(\lambda \right)$ and scattering $\beta_{s}\left(\lambda \right)$ (e.g., Rayleigh and Mie scattering) of the specific gas. Both parameters can be described with the absorption $\sigma_{a}\left(\lambda \right)$ and respectively scattering cross-section $\sigma_{s}\left(\lambda \right)$, expressed in ${\mathrm{cm}}^{2}$, and the density *N* of the medium, expressed in ${\mathrm{cm}}^{-3}$.
(40)$$\beta_{e}\left(\lambda \right)=\beta_{a}\left(\lambda \right)+\beta_{s}\left(\lambda \right)=N\;\cdot \;\left(\sigma_{a}\left(\lambda \right)+\sigma_{s}\left(\lambda \right)\right)$$

In a mixture of different gases, the extinction coefficients should be added up. A similar model can be drawn for liquids:(41)$$\eta_{link,liq}=10^{-\varepsilon cd},$$
with ε bring the molar attenuation coefficient in ${\mathrm{cm}}^{2}\;{\mathrm{mol}}^{-1}$ or $l\;{\mathrm{mol}}^{-1}\;{\mathrm{cm}}^{-1}$ and *c* the molar concentration in $\mathrm{mol}\;{\mathrm{cm}}^{-3}$.

A PV array is present on the receiving side. The theoretical model assumes that the panel can receive the entire area of the beam spot even if there is no perpendicular irradiance to the panel. The divergence of the laser beam results in an increased area of the beam spot, which is proportional to the distance from the laser. The angle of divergence of the beam $\phi_{div}$ in rad can be determined with (Equation 42) in which the laser beam profile and therefore also the divergence is considered Gaussian [54]. The beam radius at the laser is represented by *r*.
(42)$$\phi_{div}=\frac{2\lambda }{\pi r}$$

To find the diameter of the received beam *D* at a distance *d*, trigonometric functions are used:(43)D=2dtan(ϕ/2)

Following prior assumptions, the efficiency of the PV panel can be determined as follows [50]:(44)$$\eta_{pv}=\frac{P_{pv}}{P_{rec,opt}}=\eta_{cell}\left[1-\theta \left(T_{pv}-T_{ref}\right)\right],$$
with θ being a temperature coefficient for adjustment of the generated power in comparison with the thermal situation where the external quantum efficiency EQE(λ) of the material was taken. $T_{pv}$ and $T_{ref}$ are the PV panel temperature and reference temperature for the reported external quantum efficiency, respectively. $\eta_{cell}$ represents the cell efficiency of the PV material and can be determined with [50]:(45)$$\eta_{cell}=\frac{\int_{\lambda_{1}}^{\lambda_{2}}q\frac{\lambda }{h\;c}\;EQE\left(\lambda \right)\;\cdot \;FF\;\cdot \;V_{oc}\;\cdot \;I_{laser}\left(\lambda \right)}{\int_{\lambda_{1}}^{\lambda_{2}}I_{laser}\left(\lambda \right)},$$
assuming that *c* is the speed of light, FF the fill factor, $V_{oc}$ the open-circuit voltage, and $I_{laser}\left(\lambda \right)$ represents the spectral irradiance of the laser which is evaluated between wavelengths $\lambda_{1}$ and $\lambda_{2}$. The temperature has a major impact on the efficiency of the PV panel as a result of additional losses, for example for cooling, to be introduced.

At the end, the load is connected to the pv array using a power converter. Typical DC–DC converters have an efficiency $\eta_{pc_{2}}$ of about 90% [49].

The total efficiency of the LPT system can be determined as follows:(46)$$\eta_{overall}=\frac{P_{load}}{P_{psu}}=\eta_{pc_{1}}\;\eta_{laser}\;\eta_{link}\;\eta_{pv}\;\eta_{pc_{2}}$$

Due to the evolution of new structures and materials for laser and pv technologies, overall efficiencies could exceed 50%. Currently, the efficiency is being increased by the use of high-performance components, which is fundamental but corresponds with a higher cost, in particular by reducing the ripple on the driving current and by controlling the laser with a pulse current [49]. This last option makes it possible to integrate optical communication, leading to a dual-use system. However, a compromise must be made between modulation efficiency for communication and power transfer efficiency. Another advantage of an HILPB system is that it is suitable for power delivery to electronic devices in EMI, RF, high voltage and magnetic field areas [49] or for devices located underwater [55]. There are also some disadvantages such as the size, cost, and weight of the laser, the heat development which makes cooling important, and the fact that light (also in the infrared area which is used for LPT)) does not pass through walls or other objects [49]. Furthermore, there are risks related to safety, which are discussed in Section 7.

Even more research on LPT is conducted since applications such as power delivery for UVs and space applications become increasingly important [49]. A summary of some LPT realizations can be seen in Table 2 and is further discussed.

A fully autonomous rover, powered over a distance of 30– 200 m with a tracking laser having an output power of 5 W is reported in [56]. NASA, together with partners, is strongly committed to LPT research and demonstrated it over a distance of 15 m with a manually tracked laser-powered aircraft, using a 500 W laser beam. A total of 7 W of the incident laser power of 40 W was available to power the motor [57]. An LPT system for space solar power systems (SSPS) based on a solar pumped laser was proposed which delivers 1 GW in its full configuration [63]. LPT technology was applied to power a small rover, kiteplane, and drone [58]. A 60 W laser diode was used to power the rover over a distance of 1 km. The rover received a usable power of a bit more than 10 W, resulting in an efficiency of 20% without taking into account the laser efficiency. Their latest setup made it possible to fly a drone that requires 90 W of power at an altitude of 50 m. A laser output of 360 W was sufficient [59]. An LPT system with an overall efficiency of 11.6% at a distance of 100 m was demonstrated [60] with a usable received power of 9.7 W, using an optimized photovoltaic (PV) converter and a laser with a beam output power of 25 W. By using a high efficiency photovoltaic cavity converter (PVCC) [61], higher efficiencies may be achieved. Due to a bad match between the Si cells and the 1064 nm wavelength, the low flux density inside the sphere, the low reflectance of the AR coating, and the low cell population inside the cavity, a limited efficiency was achieved (<14%). With corrections, efficiencies of 40–60% are possible, according to [61]. An underwater LPT setup was tested [62], resulting in an overall efficiency of 4%. The laser output power corresponded with 50 mW. The transfer distance was only a couple of millimeters, while at a distance of e.g., 4 m the overall efficiency dropped to 0.5 to 1%. The efficiency depends on the type of water (tap water, seawater) which is not the case with air as a medium where the efficiency remains at about 4%. A very promising development is a distributed laser charging (DLC) system that generates a resonating beam between transmitter and receiver by using a gain medium at the transmitter and retroreflectors at both sides [64]. The retroreflector reflects light back to its source with three perpendicularly arranged mirrors. This gives a number of advantages. First, the resonator is formed regardless of the incident angle, with the consequence that only an LoS is needed and self-alignment is ensured. A second advantage is the intrinsic safety, since an interruption of the los immediately stops the amplification, as photons cannot resonate through obstacles. Finally, this system is able to charge multiple devices simultaneously using one DLC transmitter. Unfortunately, the efficiency, determined with the analytical model described in [65], is still rather low.

## 4. Acoustic Technologies

In acoustic power transfer, acoustic waves are used as carriers to convey energy. A typical structure of an APT system is depicted in Figure 24. In general, it consists of a pair of acoustic transducers separated by a medium. At the transmitting transducer, electrical energy is converted into vibrations, which in turn result in pressure waves radiating throughout the medium. The propagated pressure waves are then collected by a receiving transducer and converted back into electrical power. Finally, a rectifier ensures a stable DC voltage, which can be used to drive a load or charge an energy buffer (e.g., battery).

A major advantage of APT in comparison to EM-based energy transfer systems arises from the much lower propagation speed of acoustic waves with respect to electromagnetic waves. Although this velocity depends on the medium through which it is traveling, it is in general about five to six orders of magnitude smaller than the speed of light. This means that for a given wavelength, the operating frequency of the APT can be lowered by the same factor while still maintaining a comparable directionality to that of the EM system. As a result, the electronic driver circuitry can be simplified and losses can be decreased [66]. Alternatively, the designer can opt to keep the operating frequency of the APT fixed and reduce the transducer dimensions. Furthermore, APT can be used in applications where EM-based energy transfer is difficult or not an option, e.g., in the case of metal shielding.

In the remainder of this section, three main application domains of APT are discussed. They are classified according to the propagation medium, namely living tissue, metal, and air. Most research is carried out in the former two groups.

### 4.1. Biomedical

Implantable medical devices (IMDs) are placed either partly or totally into the human body and assist in the monitoring of biological parameters, drug delivery, or functional improvement of certain organs [67,68]. The main challenge in the development of these devices is miniaturization. The implants should be as small as possible to ease the surgery procedure and limit trauma to the patient. Traditionally, batteries have been a reliable power source with relatively high energy density. However, their miniaturization has not progressed at the same pace as sensing and computational components. As batteries often dominate the available space of an IMD, alternative methods are explored to power implants.

One research field focuses on wireless power transfer techniques to overcome the hurdle of battery miniaturization. As the battery of the IMD can be recharged, its size can be reduced significantly. Moreover, surgical interventions remain limited, as a replacement is only needed at the end of the battery’s lifetime. Various near-field to far-field wireless power transfer methods have been proposed for IMDs, such as inductive WPT, far-field RF WPT, and APT. However, as the size of the implant shrinks, the efficiency of inductive and RF power transfer decreases significantly due to the relatively large wavelength and attenuation at high frequencies [68]. In contrast, models have shown that APT outperforms inductive WPT when the charging distance becomes large in comparison to the implant size [69]. Moreover, acoustic waves experience less attenuation when traveling through human tissue, allowing charging at deeper penetration.

The overall power transfer efficiency of an APT system depends on various factors. A brief overview is provided below, based on the work of [1]. The first factor that influences the overall efficiency is the material of the transducer. Lead zirconate titanate (PZT) is a common piezoelectric ceramic material used for the ultrasonic transducer, as it exhibits a high electromechanical energy conversion efficiency. Another frequently used material is polyvinylidene fluoride (PVDF). This piezoelectric polymer shows better flexibility properties but has a lower electromechanical energy conversion efficiency [70].

Unfocused disc-shaped ultrasonic transducers generally exhibit a natural focusing behavior. The resulting pressure field is divided into three zones: the near field, far field, and focal zone. In the near field, the pressure field goes through a series of minima and maxima in quick succession, with the overall envelope oscillating. Consequently, power transfer in this zone is unpredictable [1]. Next, the pressure field converges and transitions into the focal zone. Eventually, the beam spreading reaches a minimum at the so-called Rayleigh distance. At this distance, acoustic pressure is at its highest level and consequently constitutes the best location for an IMD to achieve maximum received power. The Rayleigh distance L can be calculated using (Equation 47) [68].
(47)$$L=\frac{D^{2}-\lambda^{2}}{4\lambda }\approx \frac{D^{2}}{4\lambda },D^{2}\gg \lambda^{2}$$
where *D* is the aperture width of the transmitting transducer and λ the wavelength of the acoustic wave in the medium.

Beyond the self-focusing zone (far field), the pressure field transforms into a spherical spreading wave. The intensity decreases at a rate proportional to the square of the TX-RX distance, and the beam spreads out at angle $\theta_{d}$ [68]:(48)$$\theta_{d}=\sin^{-1}\left(\frac{1.22\lambda }{D}\right)$$

It is important that the APT is tuned at a proper operating frequency as it affects several other critical system parameters such as tissue attenuation, transducer size and thickness, reactive components of the transducers, and Rayleigh distance. In order to achieve maximum power transfer, the transducers must operate close to their resonance frequency, which usually ranges from a few hundred kHz up to a few MHz. However, this in turn depends on the geometry and material of the transducer. Inevitably, a trade-off must be made. On the one hand, a frequency increase results in a smaller transducer thickness and matching layer, boosting miniaturization. On the other hand, the Rayleigh distance increases with frequency, however, at the expense of increased tissue absorption and quadratic intensity decrease with distance [1,68].

An important factor in the APT efficiency is the acoustic impedance. An impedance mismatch between the transducer and tissue causes the pressure wave to be reflected back. The reflection coefficient Γ for normal incidence is given by (Equation 49).
(49)$$\Gamma =\left|\frac{Z_{2}-Z_{1}}{Z_{1}+Z_{2}}\right|=\frac{P_{r}}{P_{i}}$$
where $Z_{1}$ and $Z_{2}$ are the acoustic impedance of the tissue and transducer respectively, and $P_{i}$ and $P_{r}$ are the amplitude of incident and reflected waves. The acoustic impedance of a PZT transducer is around one order of magnitude greater than that of tissue [68]. Consequently, a large portion of the incident power, proportional to ${\left(1-\Gamma \right)}^{2}$, will be reflected back if the transducer remains improperly matched. Moreover, standing waves may occur, which can cause the pressure field to exceed tissue safety limits. In conclusion, single or multiple matching layers should be used to reduce mismatch losses [71].

Power transfer efficiencies of APT for implantable devices have been reported to range from a few percent up to around 50% [68]. However, the authors of this work indicated that APT systems with smaller receivers tend to have a lower power transfer efficiency. When the efficiencies are normalized in the function of the transducer areas, the trend of high efficiency due to large transducer area cancels. This indicates that the efficiencies in most works were affected due to unequal size matches rather than through the performance of the receiver.

Several models have been proposed to estimate the theoretical power transfer efficiency of an APT system. An electromechanical model presented in the context of millimeter implantable devices has been presented in [72]. Herein, the overall power transfer efficiency is given by:(50)$$\eta =\left|\frac{P_{out}}{P_{in}}\right|=\frac{{\left(\mu T\;\cdot \;\phi_{R}\;\cdot \;Z_{L}\right)}^{2}}{C_{TX}C_{RX}V_{IN}^{2}{\left(Z_{L}+Z_{OUT}\right)}^{2}}$$
where $P_{out}$ is the electrical output power delivered to the load, $P_{in}$ is the electrical input power, $\mu =e^{-2\alpha x}$ is the tissue attenuation factor, where α is the attenuation coefficient and *x* is the depth of the implant, $C_{TX}$ and $C_{RX}$ are the capacitance of the transmitter and receiver transducer respectively, $\phi_{R}$ is the electromechanical transformer ratio, $Z_{OUT}\left(=1/j\omega C_{RX}\right)$ is the output impedance, $Z_{L}$ is the electrical load impedance at the receiver, $T\approx 2c|Z_{\mathrm{receiver}}|$, where *c* is the acoustic velocity in the piezoelectric material and $Z_{\mathrm{receiver}}$ is the acoustic impedance of the piezoelectric material at the receiver, and $V_{IN}$ is the voltage applied over the transmitter’s capacitance.

### 4.2. Metal Wall

There exist many situations where sensors are enclosed in a metal structure or are isolated from an operator by means of a metal wall, e.g., in gas cylinders, vacuum chambers, pipelines, etc. While wall penetrations allow for easy wire feed through, they may have a major impact on the integrity of the overall structure. After all, holes form a weak spot which may increase design complexity and costs to take into account the risk for leakage of chemicals and gasses, loss in pressure or vacuum, or breakage in thermal or electrical insulation [73]. Sensors can also be encapsulated in metal walls for the benefit of structural health monitoring (SHM) [74] to monitor structural parameters such as strain, acceleration, and temperature in situ through non-destructive evaluation (NDE).

Metal walls, however, form a challenging environment for electromagnetic-based wireless power transfer. Some power can be transferred through coupling systems, but the effectiveness strongly decreases for ferromagnetic or thick non-ferromagnetic metallic barriers due to strong Faraday shielding [75]. Acoustic power transfer (APT), on the other hand, is not inhibited by electromagnetic shielding and can achieve good efficiencies and power levels. For example, in [76], 50 W AC power is transferred at an efficiency of 51% through a 63.5 mm thick steel barrier and in [77], more than 1 kW was delivered at 84% efficiency by means of a prestressed piezo actuator. Moreover, it is easier to achieve a high output power level and efficiency in the case of metallic media than through air or tissue [66]. Since piezoceramic transducers and for example, steel has a similar acoustic impedance (30 MRayl and 45 MRayl respectively), better impedance matching is achieved.

First work in the derivation of an analytic model for a planar through-wall configuration APT was reported in [78]. The efficiency of the acoustic-electric channel was calculated using the wave equation and linear equations of piezoelectricity [73]. However, only thickness vibration modes were assumed, and also bonding layers were neglected. Later on, cylindrical configurations and nonlinear effects were also studied using the same coupled continuum approach [79]. A disadvantage of these models resides in the bulky elaboration and evaluation in the case of channels with many layers [79]. Alternatively, models based on equivalent circuit modeling have been worked out, which can be easily adapted to fit additional acoustic elements or loss mechanisms. Moreover, they can be connected directly to other networks such as power processing circuitry (diode bridge, capacitors, rectifiers, etc.) [73]. Two commonly used equivalent circuit models are the Mason [80] and KLM model [81]. Both the coupled continuum and equivalent circuit model approach are actually different implementations of the one dimension propagation model (ODPM). Throughout the years, new implementations have been proposed that allow simulation of complex systems with good approximation, e.g., [82]. Finite element analysis offers the most comprehensive model and allows for complex geometries, yet is rather computationally intensive. When the degrees of freedom becomes too high (e.g., high-frequency and/or large geometry models), the method becomes impractical [79].

### 4.3. Air

The research on APT in gaseous media such as air is far more limited than the biomedical and through-wall APT research areas. Nonetheless, wireless power transfer through airborne ultrasound can be advantageous over electromagnetic-based WPT in some cases. As mentioned before, the propagation speed of acoustic waves $c_{air}$ in air is much lower than that of electromagnetic waves $c_{EM}$. Consequently, the dimensions of the transmitter and receiver can be a factor $c_{EM}/c_{air}$ smaller than their EM-based counterparts for a given directionality [66]. This can, for example, be a decisive factor in the context of Internet of Things (IoT) where preferably small sensors are deployed. Moreover, APT can be used in environments where strong EM fields must be avoided due to health and safety issues or EM propagation is complicated by the presence of metallic objects [83].

Acoustic power transfer (APT) through air comes with a number of drawbacks. Since acoustic waves have small wavelengths in air, diffraction around obstacles is rather unlikely. Consequently, the transmitter and receiver must preferably be positioned in the line of sight to obtain reasonable efficiency. Furthermore, acoustic waves are more subject to absorption in air than electromagnetic waves, respectively in the order of a few ${\mathrm{dBm}}^{-1}$ versus decibel/km [84]. Moreover, distortion due to the nonlinear pressure–density relation of acoustic media already occuring at intensities below the regulation limits. This causes an energy shift towards the harmonics, where a higher absorption leads to increased dissipation losses [84].

In conclusion, APT is an indispensable technology in applications where device miniaturization is important and EM-based WPT is difficult. Depending on the propagation medium, power levels are restricted due to health implications.

## 5. Range, Power, and Efficiency-Increasing Technologies

This section discusses techniques that can increase the range, power, and/or efficiency of a wireless power transfer system. The main strategies considered are beamforming, repeaters, power transfer through uvs, and medium optimization.

### 5.1. Beamforming

A well-known approach to improving the performance of RFPT systems is phased array transmission. In this case, multiple antenna elements are arranged in an adequate configuration and excited such that the individual fields constructively combine at the receiver device [85,86,87]. The resulting array system offers several advantages over a single antenna setup. First, more power can be transferred to the receiver given the increase in antenna gain, also known as the array gain. Second, the beam pattern can be adjusted according to the situation. For example, when no los path is present between transmitter and receiver, the beam pattern can be modified to maximize power transfer. To be complete, the same principle can be applied to the receiver [88,89,90]. However, this is not commonly done as the addition of the necessary circuitry and processing, increases the cost, complexity, form factor, and energy consumption of the receiver. Beamforming comes with a number of challenges. In many situations, both the transmitter and receiver do not know their relative position to each other ahead of time [86,91]. Consequently, they are not able to adapt their beam patterns to maximize received power immediately. This becomes even more challenging for mobile receiver devices. A fast beam steering and/or focusing algorithm is thus necessary to optimize the power transfer in real-time. However, this may be complicated. Passive devices, a group that could significantly benefit from beamsteering, might not have the necessary energy budget to achieve rapid adjustment. Additionally, obstructions, reflections, and/or scattering in indoor environments can make it hard to find the optimum beam. In [92], an alternative approach is introduced that takes the advantage of an array setup, but greatly reduces complexity. In normal receive beamforming, the signals from all antenna elements are combined before they are passed to a single rectifier. However, in this work, each antenna signal is sent to a separate rectifier after which the DC powers are combined. Regular receive beamforming requires channel state information (CSI) at both the transmitter and receiver for beamforming optimization, while only the CSI at the transmitter is necessary in the latter case. While this significantly reduces complexity, a higher efficiency can still be obtained through regular receive beamforming due to the rectenna nonlinearity. Beamforming is not restricted to the domain of RF. For example, in the domain of acoustics, phased arrays have been playing a leading role in the context of noninvasive diagnostic examination in medical applications, nondestructive testing, and sonar. However, it was only until recently that an ultrasound phased array system has been used to enhance the performance of through-air acoustic WPT [93]. Additionally, in the domain of magnetic resonance coupling, beamforming has been introduced to increase the power transfer efficiency [94,95,96,97]. One could argue that laser-based power transfer which happens in a focused beam in fact also implements a beamsteering approach.

### 5.2. Repeaters

The operating distance can be increased by applying repeaters between the transmitter and receiver. For example, MRC and CPT coverage extension can be achieved by introducing additional resonance tanks. Similarly, Wan et al. [98] have shown that RF repeaters can be efficiently used to extend the range. However, we do not treat these RF extensions here as these require extra hardware between transmitter and receiver, which may not be a practical solution in every application to increase coverage. We further elaborate on the magnetic field and electric field repeaters below.

#### 5.2.1. Magnetic Field Repeater

As discussed in Section 2, typical IPT systems use tightly coupled coils to achieve high-efficient WPT. However, these systems require a short distance. To extend the distance, MRC systems were designed using loosely coupled coils. A repeater circuit, consisting of a capacitor $C_{R}$ and inductor $L_{R}$, can be used to further increase the Tx-Rx distance. Figure 25 shows the MRC scheme with an LC repeater.

The equivalent series resistances for the coils $L_{1}$, $L_{R}$, and $L_{2}$ are $R_{L_{1}}$, $R_{L_{R}}$, and $R_{L_{2}}$. Adequate capacitors $C_{1}$, $C_{R}$, and $C_{2}$ ensure that all circuits operate in resonance (Equation 51). $k_{1}$ and $k_{2}$ represents the coupling factors between $L_{1}$ and $L_{R}$ and between $L_{R}$ and $L_{2}$ respectively.
(51)$$\omega_{0}=\frac{1}{\sqrt{L_{1}C_{1}}}=\frac{1}{\sqrt{L_{R}C_{R}}}=\frac{1}{\sqrt{L_{2}C_{2}}}$$

Applying two times the equivalent T-model circuit results in the scheme in Figure 26. Using KVL gives (Equation 52).
(52)$$\begin{array}{c} \left(R_{S}+R_{L_{1}}\right)I_{1}-j\omega M_{1}I_{R}=V_{S}\\ j\omega M_{1}I_{1}-R_{L_{R}}I_{R}+j\omega M_{2}I_{2}=0\\ -j\omega I_{R}+\left(R_{L_{2}}+R_{L}\right)I_{2}=0 \end{array}$$

The resistive contributions from transmitter $R_{TX}$, repeater $R_{R}$ and receiver $R_{RX}$ are listed in (Equation 53).
(53)$$\begin{array}{c} R_{TX}=R_{S}+R_{L_{1}},\;R_{RX}=R_{L_{2}}+R_{L},\;R_{REP}=R_{L_{R}} \end{array}$$

The transmitter, repeater, and receiver quality factors are $Q_{TX}$, $Q_{R}$, and $Q_{RX}$ respectively. Solving (Equation 52) gives the expression for the receiver current in (Equation 54).
(54)$$I_{2}=\frac{\omega_{0}\sqrt{L_{1}L_{2}}/\left(R_{TX}R_{RX}\right)}{\frac{k_{1}}{k_{2}}Q_{TX}+\frac{k_{2}}{k_{1}}Q_{RX}+\frac{1}{Q_{R}k_{1}k_{2}}}V_{S}$$

The denominator will be substantially smaller compared to the denominator from (??), which represents the receiver currently in an MRC system without a repeater. Here, the term $1/(Q_{R}k_{1}k_{2})$ is almost negligible, especially when the quality factor from the repeater is high. The overall efficiency of such an approach is given by (Equation 55) with the current ratios shown in (Equation 56) and (Equation 57).
(55)$$\eta_{overall}=\frac{R_{L}}{R_{TX}{\left|\frac{I_{1}}{I_{2}}\right|}^{2}+R_{REP}{\left|\frac{I_{R}}{I_{2}}\right|}^{2}+R_{RX}}$$
(56)$$\frac{I_{1}}{I_{2}}=-\sqrt{\frac{L_{2}}{L_{1}}}\frac{1}{k_{1}}\left(\frac{1}{k_{2}Q_{R}Q_{RX}}+k_{2}\right)$$
(57)$$\frac{I_{R}}{I_{2}}=-\frac{R_{RX}}{jk_{2}\omega_{0}\sqrt{L_{2}L_{R}}}$$

To achieve high efficiencies, the current ratio $I_{1}/I_{2}$ should be as small as possible. This ratio can be minimized by increasing $k_{1}$ and thus placing the repeater close to the transmitter. However, the placement of the repeater near the transmitter is limited by the term $1/(k_{2}Q_{R}Q_{RX})$. The current ratio from (Equation 57) could also be minimized, hence increasing $k_{2}$. Knowing that the power loss in the transmitter is typically higher than the loss in the repeater ($R_{TX}>R_{L_{2}}$), overall the efficiency increases by placing the repeater slightly closer to the transmitter. The optimal position depends on the optimal coupling factor $k_{2}$ for a given $k_{1}$, shown in (Equation 58) [10].
(58)$$k_{2}={\left(\frac{1+k_{1}Q_{R}Q_{TX}}{Q_{R}^{2}Q_{RX}^{2}}\right)}^{0.25}$$

The concept can be extended to multiple repeaters. While this requires attention regarding alignment, experiments have shown a potential. Kurs et al. [99] use self-resonant coils operating at 9.9 MHz. They transmitted 60 W over a distance of 2 m, which is eight times the radius of the coils they used, with an overall efficiency of 40–50%.

#### 5.2.2. Electric Field Repeater

The biggest challenge in CPT is to overcome the low coupling. As a result, power transfer is only feasible for distances up to ten cm. To achieve larger distances, an electric field repeater is required, an example is shown in Figure 27 [100]. Additional repeater plates with LC circuits are added to the four-plate structure. These plates are not galvanically connected to either the transmitter or the receiver. For larger distances, more repeaters can be added. Each repeater segment introduces additional power losses, thus maintaining efficiency forms the main challenge [100,101]. Similar to the magnetic field repeater, the formulas for the overall efficiency can be derived by using an equivalent scheme for the capacitive coupler. A reported experiment shows that, with one electric repeater, 100 W can be transferred over 36 cm at an efficiency of 66% [100].

### 5.3. Energy Sources Carried by Unmanned Vehicles (UVs)

Conceptually, moving energy “wirelessly” from place A to B can be considered WPT. Thus, energy transfer via a uv is also WPT, certainly, when the actual delivery happens without wired contact, through technologies discussed in Section 2, Section 3, Section 4. Batteries, effectively, become mobile by the uv and are transported to places where energy is required. To realize this concept, a WPT-enabled UV is introduced that is able to move through the air, on the ground, or underwater, i.e., through an unmanned aerial vehicle (UAV), unmanned ground vehicle (UGV), or unmanned underwater vehicle (UUV) respectively. The main benefit of employing a UV is to charge sensor modules remotely and thereby ensure the longest possible operation without human intervention. In the remainder of this section, several types of UV-supported WPT and associated challenges are discussed. Underwater sensors can be recharged autonomously with UUVs. The main challenge resides in the influence of the medium. For example, the widely used IPT technology has a reduced efficiency in salt water compared to air [102]. A UGV equipped with, e.g., mecanum or omnidirectional wheels, can serve as a flexible solution for autonomously recharging devices located on the ground. The vehicle has an amount of stored energy available and can be used in residential or industrial applications [103]. A UAV provides an interesting vehicle for recharging distributed battery powered IoT nodes. Current nodes typically have a limited autonomy of months to a few years only. Consequently, the adoption of wireless power receivers in future nodes should make it viable to increase the autonomy significantly [104]. Regardless of the type of UV, localization and alignment is one of the major challenges since efficient coupled WPT links operate over small distances. The associated requirements are linked to the technologies, e.g., IPT, MRC, CPT. If smaller amounts of energy suffice, RF-based power transfer can be realized from a UV at larger distances. Consequently, alignment is no longer a challenge. The RF link can energize the sensor nodes and the UV can receive its data via backscattering [105].

### 5.4. Favorable Propagation Medium

Surrounding elements in a WPT system can be exploited to improve the system performance. However, these elements can also cause severe degradation due to induced eddy currents, blocking the line of sight or attenuating EM signals. In Section 3.2, we already mentioned that fog or rainy weather attenuates the light beam, resulting in an efficiency reduction. The wavelength plays an important role in RFPT systems, especially when there are objects between the transmitter and receiver. Higher frequency em waves experience difficulties passing through objects, walls, etc. Due to the restrictions in maximum transmit power, RFPT transmitters rely on los, otherwise, the received power is too low. The material properties, permeability, and permittivity affect IPT and CPT systems as well, which therefore requires some attention during the design.

#### 5.4.1. Inductive Power Transfer

In particular, the permeability μ strongly influences the transfer, since materials with higher permeability concentrate the magnetic fields. The relative permeability $\mu_{r}$ (referred to $\mu_{0}$) of wood and water, for example, is almost the same as $\mu_{0}$. Such materials have little influence on the system performance, and the magnetic field lines will easily pass through. Instead, for iron or ferrite materials, the permeability may be a thousand times higher than $\mu_{0}$. Since such materials conduct the field lines better, they can influence inductively coupled systems. In low-frequency IPT, the transmitter and receiver coil are typically surrounded by flexible absorbent sheets or ferrite plates to minimize interference with the underlying circuits. Bringing these materials into the magnetic field results in losses within the magnetic materials due to hysteresis and eddy currents. To gain a better understanding of the losses, the relative permeability is separated into an ideal part $\mu^{\prime }$ and the losses or reactive part $\mu^{\prime \prime }$ represented as $\mu_{r}=\mu^{\prime }-j\mu^{\prime \prime }$. The approach to concentrate the magnetic field and protect the surrounding circuits is to use materials with high $\mu^{\prime }$ and low $\mu^{\prime \prime }$ [106].

#### 5.4.2. Capacitive Power Transfer

CPT is a technology that can transfer energy over a few millimeters and by using the right compensation circuits up to a dozen of centimeters [15,16,18,19,100]. This is mainly true for air but what about other media? According to (Equation 59), to calculate the capacitance of a capacitor, the medium between the plates has a major impact on the capacitive coupling.
(59)$$C=\varepsilon_{0}\;\cdot \;\varepsilon_{R}\;\cdot \;\frac{A}{d}$$

The main capacity between two plates can be determined by the vacuum permittivity which is constant $\varepsilon_{0}$, the relative permittivity of the medium $\varepsilon_{R}$, the surface area of the plates *A* and the distance between the plates *d*. From (Equation 59), it can be seen that for media with a higher relative permittivity than air, the resulting capacity will also be proportionally larger. As an example, four square plates with a side of twenty centimeters are taken. The distance between transmitter and receiver is one centimeter, so the parasitic capacities can be neglected. Table 3 compares this setup for different materials and shows the potential positive effect. Thus, in theory, a different medium can enable capacitive power transfer over longer distances without using complicated compensation circuits. More research on the influence of a medium is needed to open opportunities for actual applications [107].

## 6. Standards and Commercial Solutions

This section presents a comprehensive overview of current WPT standards, including proprietary implementations designed due to the lack of standards for specific use cases. First, the inductively coupled systems, more precisely the IPT and MRC specifications are discussed. Secondly, the CPT based implementations will show the potential of electric field power transfer in real-life applications. Thirdly, commercial technologies and emerging systems for RFPT are handheld. Lastly, the light- and laser-based WPT implementations show that this technology has a future potential for residential and industrial applications.

### 6.1. Inductive and Magnetic Resonance Coupling

This section discusses inductive coupling standards and implementations. The distance between the transmitter and receiver coil is fairly small, typically from one to a few centimeters. The frequency band for these systems ranges from 50 kHz to 13.56 MHz.

#### 6.1.1. Wireless Power Consortium

The The Wireless Power Consortium (WPC) is an open, collaborative standards development group with more than 400 member companies from around the globe. It provides a wide range of power transmission systems. They distinguish four standards: the Qi standard, Ki Cordless Kitchen standard, light electric vehicle (LEV) standard, and Industry standard.

**Qi standard** is based on inductive coupling with tightly coupled coils, meaning that the spatial freedom is low when charging devices. The charging efficiency reaches values above 70%. A bidirectional communication link is established using load modulation and frequency-shift keying (FSK), thereby omitting the need for a supplementary radio system. The standard is developed for smartphones and other mobile devices and can transfer up to 30 W [2]. Future extension of the specification will deliver up to 60 W to enable, e.g., laptop charging [109]. It also features halting the WPT when efficiency drops due to, e.g., misalignment, and when foreign objects, e.g., metals, are detected potentially causing safety concerns.

**KI Cordless Kitchen standard** is currently under development with the purpose to power kitchen appliances up to 2200 W. Examples like rice cookers, toasters, blenders, coffee makers, air fryers, and more can be powered wirelessly making the cord no longer necessary. A first draft specification is available for members of the Wireless Power Consortium [109].

**LEV standard** under development describes the specifications to charge electric bikes and scooters, complementing existing standards designed for electric vehicles, e.g., cars. LEV charging is currently implemented with proprietary implementations. The WPC wants to develop the LEV standard to ensure interoperability by working with industrial partners [109].

**Industry standard** should enable safe wireless charging of industrial battery-powered vehicles in the future. There are already companies whose UGVs-are wirelessly recharged. As a result of a missing industrial WPT standard, there is currently no interoperability between existing systems. WPC wants to change this by working with industry partners and developing their Industry standard [109].

#### 6.1.2. AirFuel (Alliance) Resonant

AirFuel Alliance is a merger of two prior standards groups, PMA and A4WP. The organization focuses on the creation of standards based on magnetic resonance and RF wireless power. The IEC 63028 *Airfuel Alliance resonant baseline system specification* describes the technical requirements, behaviors, and interfaces used for ensuring interoperability for loosely coupled WPT. This standard describes WPT via loosely coupled coils driven by a 6.78 MHz frequency. The transmitter and receiver are called, respectively power transmitting unit (PTU) and power receiving unit (PRU), as shown in Figure 28. Due to resonance coupling, a device can be powered up to 50 mm away from the transmitter, meaning that the operation range is larger compared to Qi-enabled appliances. The PTUs and PRUs are subsequently divided into classes and categories, each having a distinct maximum receive power (PRU) and minimum charging area (PTU). The maximum receive power ranges from 1.5 to 50 W and the minimum antenna size from 50 × 50 to 120 × 110 mm^2^. AirFuel Alliance uses Bluetooth Low Energy (BLE) to communicate between the receiver and transmitter, in contrast to the Qi standard, where communication happens through the coupled coils. The key benefits are spatial freedom and multi-device charging. Moreover, these systems can be mounted under a desk or table, making the installation more practical. AirFuel Resonance could possibly become Qi’s main competitor, although currently, there are not many devices that support this standard [110].

#### 6.1.3. Wireless Charging Specification (WLC)

The wireless charging specification (WLC) standard created by the NFC forum describes how to charge small, battery-powered consumer electronics or IoT devices with a smartphone. It makes use of MRC. The wlc enables both communication and charging with an energy transfer rate categorized into four power classes: 250, 500, 750, and 1000 mW. These upgrades are promising to charge devices such as smartwatches, wireless earbuds, etc. During energy exchange, the 106 kbps data throughput remains fairly high compared to the maximum 424 kbps in the NFC standard.

#### 6.1.4. Standards for Automotive

The number of electric vehicles (EVs) and charging stations is rapidly increasing. A lot of standards have been developed for EVs wired charging. More on these can be found in [111]. We here below discuss wireless charging options.

**SAE J2954** is the US standard for wireless charging EVs and describes the *Wireless Power Transfer for Light-Duty Plug-in/Electric Vehicles and Alignment Methodology*. This standard is subdivided into three plug-in hybrid electric vehicles (PHEV) classes, 3.7 kW, 7.7 kW, and 11 kW. Resonant inductive coupling is used, operating in a frequency range of 81.38– ground assembly (GA)90 kHz ( 85 kHz) [112]. Figure 29 illustrates the system architecture. The charging station and the vehicle-mounted system are called, respectively, ground assembly (GA) and vehicle assembly (VA). Wireless energy transfer requires communication between EV supply equipment and plug-in electric vehicle (PEV). The SAE J2931/1 describes digital communication, primarily for wired charging (SAE J1772). The SAE J2847/6 covers the requirements specifically for the Communication Between Light-Duty Plug-in Electric Vehicles and Wireless EV Charging Stations. The communication between GA and VA uses an IEEE 802.11n (Wi-Fi) interface. The new SAE J2954/2 standard under development will define *Wireless Power Transfer and the Alignment for Heavy Duty Applications*. This guideline will cover criteria for interoperability, electromagnetic compatibility, minimum performance, safety, etc. SAE J2954 is meant to harmonize with standards developing organizations in order to make a world-wide WPT standard up to 500 kW [113].

EVs can charge as fast and as efficiently as conventional PEV systems. WiTricity is a technology supplier whose systems are compatible with the SAE J2954 standard. Their system is equipped with foreign object detection (FOD), live object detection (LOD) and position detection (PD) and can reach charge distances up to 10–25 cm. They claim the option of bidirectional power transfer, making it possible to use EV batteries to stabilize the grid or power a home [114,115]. Other OEMs and technology suppliers include Delpi, Lear, LG, Magna, Panasonic, and TDK [116].

**IEC 61980** is the international standard for WPT to EVs and is divided in three parts. IEC 61980-1 contains general requirements and covers the operating conditions, safety and electromagnetic compatibility (EMC) requirements of the supply device, the communication between EV device and vehicle to control WPT, the efficiency and alignment. IEC 61980-2 defines the communication between the EV and the WPT infrastructure. IEC 61980-3 elaborates on magnetic field WPT and includes the operating conditions, the electrical safety, the basic communication, the requirements for positioning to assure efficiency and safe power transfer, and the EMC requirements. The International Organization for Standardization (ISO) describes similar parameters in the **ISO 19363** standard. The operating frequency and power classes are similar to the SAEJ2954 standard. ISO/IEC and SAE are working with close cooperation for harmonization [117].

#### 6.1.5. Proprietary Solutions

Several companies offer solutions for industrial applications, mostly with proprietary implementations. They conform with the safety regulation discussed in Section 7, but otherwise have no specific documentation available to develop compatible devices.

**LinkCharge** is developed by Semtech and provides wireless power transmitter and receiver solutions on the market that can deliver up to 40 W. They categorize these standards as low, medium, and high power. Evaluation boards for each power subcategory are available. Most of the evaluation boards supporting the LinkCharge medium and high power also support the Qi standard, allowing charging of both LinkCharge and Qi receivers [118].

**IN${}^{2}$POWER** is a Belgian subsidiary of the engineering firm Inverto, responsible for commercializing and manufacturing WPT products. IN${}^{2}$POWER has marketed the iN.CHARGE, a 16 kW wireless inductive charging system with a reported efficiency of 95% over a charging distance between 1 and 5 cm. The secondary side (receiver), connected to a battery, can handle voltages up to 120 V and deliver currents up to 250 A [119]. Devices can be installed in parallel with each other, resulting in power levels up to 48 kW [119]. This system is not compatible with any current wireless charging standard due to its high power level, whereby IN${}^{2}$POWER always delivers the transmitter as well as the receiver. iN.CHARGE mostly targets the automated guided vehicle (AGV) market and the logistic, robotic, medical, and nautical sectors.

### 6.2. Capacitive Coupled Systems

#### 6.2.1. ARIB Standard

The Association of Radio Industries and Businesses (ARIB) defines wireless power transmission systems in the ARIB STD-T113 standard. This standard is divided into three parts, one of them is about capacitive coupling wireless power transmission systems for mobile devices. The scope of this standard is given in Figure 30. It specifies a 400 kHz wireless interface between a power transmitting unit (PTU) and power receiving unit (PRU) and describes its operating conditions, system parameters, electrode design, system control requirements, and emc requirements [120].

#### 6.2.2. Murata

Murata developed the capacitive power transfer module LXWS series, which can charge 10 W and was mass-produced as an iPad2 accessory in 2011. Murata also gave a number of demonstrations, e.g., at CES 2012 [121]. Unfortunately, this product is no longer on the market today.

#### 6.2.3. Eggtronic

Eggtronic is an Italian power electronics company founded in 2012 [122], specializing in power converters and wireless power transfer solutions. At CES 2019, they demonstrated a moving object on a rail powered by capacitive power transfer. This demonstrator featured a 100 W vehicle lamp equipped with a limited capacitive coupling surface. This demonstrated a large positional freedom and high power density over the surface [122]. Eggtronic claims to have R&D prototypes that are able to power smartphones, laptops, and TVs via CPT.

#### 6.2.4. Solace Power

Solace power is a company that specialized in wireless power transfer. They provide a product platform named Equus. This is a patented resonant capacitive coupling system which can deliver 250 W across 375 mm [123].

### 6.3. Radio Frequency Power Transfer Systems

In this section uncoupled wireless energy transfer implementations are discussed for radio frequency, ultrasound, and laser light-based applications.

#### 6.3.1. Airfuel RF (Alliance)

AirFuel Alliance, as mentioned, besides Magnetic Resonance also covers RF wireless power. AirFuel RF uses radio waves to transmit power from an RF transmitter to an embedded device. A low amount of energy can be delivered to wearables, medical devices, etc. at distances ranging from a few centimeters to a meter. This technology benefits devices that require spatial freedom. e.g., the company Energous Corporation implements the Airfuel RF technology under the product name WattUp. Dialog Semiconductor produces the WattUp energy harvesting ASICs. DA2210 and DA2223 are two WattUp-compatible receiver ICs. Depending on the application, the developer can connect up to four antennas. The DA4100 is an example of a WattUp-compatible transmitter IC. A schematic representation is shown in Figure 31.

#### 6.3.2. Ossia Inc. and ARCHOS

Ossia Inc. and ARCHOS announced in 2021 their partnership to launch wirelessly powered products (e.g., indoor cameras, air quality and temperature sensors, smart trackers, smart health watches) that utilize Ossia Inc.’s Cota RFPT-based system. A showcase of the Cota-enabled ARCHOS devices took place at CES 2022 [125]. The technology recently received an FCC, CE, and UK certificate and is therefore a pioneer in bringing RFPT to the market. They claim to be inherently safe, without a necessary exclusion zone in front of the transmitter and without the need for power transfer interruption during passages [126]. So far, not much info is released about exact power values and distances. The technology uses the 2.4 GHz and 5.8 GHz frequency bands, depending on the type of device (e.g., Cota Forver Tracker and Cota Home, respectively).

#### 6.3.3. Semiconductor Manufacturers

**E-peas** focuses on the development of energy management solutions to enable energy-neutral and autonomous operation of devices. They present a spectrum of energy managers capable of harvesting energy from photovoltaic, thermal, vibration, and RF sources. In the case of RF, their devices are able to harvest energy at input powers in the range of −19 dBm to 10 dBm in the 868 MHz, 915 MHz and 2.4 GHz frequency bands with an overall conversion efficiency between 10% and 60% depending on the frequency and input power, and thus distance between the transmitter(s) and receiver device.

**Powercast** also works on long-range wireless power transfer through RF. They develop several products, going from battery-free temperature scanning systems to UHF RFID batteryless retail price tags and RF energy harvester chips. They present two types of RF-to-DC energy harvester chips: (i) configurable to harvest energy in the 10 MHz to 6 GHz frequency range and (ii) capable of harvesting RF energy specifically in the 850 MHz to 950 MHz frequency band.

#### 6.3.4. EMROD

The start-up EMROD from New Zealand differs from the above as it targets a much higher power transfer. It has developed a long-range, high-power WPT system to replace the contemporary copper line technology. Due to the collaboration with Powereco, New Zealand’s second-largest electricity distribution company, this technology can get adopted more quickly. The technology is based on the use of electromagnetic waves, situated in the microwaves ism band, which covers several smaller frequency bands within the range from 6.765 MHz to 246 GHz [127]. A power beam is directed to one point, with no radiation around the beam. As an extra safety consideration, a laser grid is added to shut down the microwave link if something passes through it. They claim to obtain a system efficiency of about 70%. The small prototype of EMROD sends a few watts over a distance of around 40 m. The new prototype for Powereco targets delivery of a few kilowatts and scale up (even for 100 times more power over much larger distances). The size of the antenna and the distance has a big impact on the power transfer parameters [128].

### 6.4. Light or Lasers Power Transfer Systems

There are no standards for LPT yet. Nevertheless, research demonstrations exist, as listed in Section 3.2. A selection of the most advanced demonstrations and first commercial products are discussed here.

#### 6.4.1. WiCharge

WiCharge is a company that is able to deliver over-the-air wireless energy up to 2.25 W, using infrared (IR) light over a distance up to 10 m. They use point-to-point energy delivery, meaning that almost 100% of the optical energy from the transmitter reaches the receiver. The system is based on the DLC architecture, described in Section 3.2 [64]. Class 1 lasers suffice, which makes their implementation safe. The laser beams do not reach humans because the light beams go directly to the receivers. During energy transfer, a line of sight is required. The system is able to transfer energy even while the receiver is moving. In addition, several receivers can be targeted through multiple beams simultaneously. This company already received safety certification for the use of this system in Europe [129].

An external receiver, as small as a USB stick, can receive the IR beam and convert this light into energy with a total delivered power up to 1.5 W. Different devices can be supplied with the micro USB output. Integrated objects can also be equipped with a WiCharge receiver such as an automatic flush valve, Qi charger, power bank, smart door lock, trains for children, game equipment, etc.

#### 6.4.2. PowerLight Techologies

PowerLight Technologies, the former LaserMotive, is specialized in transferring energy with laser light. The company focuses on LPT applications such as wireless power beaming for 5G infrastructure, autonomous vehicles, temporary power delivery for example in disaster areas, military applications, smart grid, and clean energy applications, and space solutions for power distribution to lunar infrastructures. They claim to reach an efficiency of 20% to 25% on their newest technologies [130]. In 2009, they demonstrated a robotic climber that climbs to a height of 1 km harvesting over 1 kW of optical power. Later they powered a Stalker UAV and increased the flight time by 24 times using the LPT system. Although there has been great progress, relevant technical details have not been published yet [49]. A more recent demonstration was conducted with energy transfer up to 400 W output power with an input power of 2 kW, over a distance of 325 m. The photo voltaic receiver used is similar to solar cells, only these cells are tuned to the lasers’ wavelength. IR light is used in this setup. During the demonstration, a light, a laptop, and a coffee machine are supplied with power, whereby this demonstration best describes the household situation [131].

## 7. Safety and Regulatory Context

When transferring power “freely” over the air, it is crucial to ensure safe operation. Moreover, coexistence with other systems should be guaranteed and follow regulations. Those aspects are discussed in this section for electrical, magnetic, and electromagnetic fields, acoustic and IR waves respectively.

### 7.1. Exposure to Electric Fields

Electric fields can have a significant impact on the human body. For this reason, the electric field strength for a system operating at 1 MHz is limited to 614 V/m according to standard C95.1 [132]. To transfer power with CPT, strong electric fields are required. The authors of [23] showed through simulations that the electric field strength between the plates of a 2 kW system can reach 180 kV/m, which is considerably above the allowed 614 V/m in the IEEE standard. For this reason, safety environments have to be provided around such systems, in this case, the safety distance would be 1 m. By using a six-plate structure, the spread of electric fields can be limited and therefore also the safety distance.

Another concern is the high plate voltages required to generate strong electric fields. These can be several kV through the compensation networks. When a human body comes in contact with this, it can be lethal. Therefore, an insulating coating will be applied to the plates [15,133].

### 7.2. Exposure to Magnetic Fields

The body exposure limits in unrestricted environments for magnetic fields can similarly be found in standard C95.1 [132]. Inductively coupled systems should not induce a magnetic field that can reach persons above 163 A/m in the frequency range of 3.35 × 10^3^
Hz to 5 × 10^6^
Hz. MRC power transfer works with higher frequencies. In the range 1.34 MHz to 30 MHz, the magnetic field strength may not exceed $\left(16.3/f_{M}\right)$ A/m with $f_{M}$ the frequency in MHz.

### 7.3. Exposure to Electromagnetic Fields

Several organizations and governments have developed standards for exposure to electromagnetic radiation. These guidelines are based on scientific studies and can be transferred into legal regulations by each country. In the United States, the Federal Communications Commission (FCC) adopted the recommendations of two expert organizations, namely the National Council on Radiation Protection and Measurements (NCRP) and Institute of Electrical and Electronics Engineers (IEEE). Many European countries follow the guidelines developed by the International Commission on Non-Ionizing Radiation Protection (ICNIRP). The safety limits of the ICNIRP correspond almost completely with those of the NCRP and IEEE, with some exceptions [134].

A distinction in the guidelines is made based on the subject to whom the exposure applies, namely the occupationally exposed individual or member of the general public [135]. Occupationally-exposed individuals are expected to be trained for potential RF risks and harm-mitigation measures. In contrast, the general public consists of individuals of all ages, with possibly diverse health risks. It cannot be expected that they have the knowledge or are able to control their exposure to electromagnetic fields. Consequently, lower exposure limits are maintained. The permissible exposure limits for the general public in the function of the ism bands can be found in [132,136].

Two main biological effects have been identified in the scientific literature [135]:Thermal effects: heating of the biological tissue and overall increase in body temperature at frequencies >100kHz.Non-thermal effects: nerve stimulation for frequencies up to 10 MHz.

As the frequency increases, heating effects become the dominating effect, and the probability of nerve stimulation decreases. Several dosimetric quantities are used to describe the exposure limits, depending on the frequency or duration of the exposure. For example, heating effects at frequencies below 6 GHz are often specified in terms of specific energy absorption rate (SAR), while absorbed power density is regularly utilized for frequencies above 6 GHz [135]. Different values are recommended for different parts of the body and whether the source is operated close to the body or at a large distance [137].

### 7.4. Exposure to Ultrasound

#### 7.4.1. Tissue

The potential bioeffects of ultrasound in tissue are generally classified into two groups: the thermal and non-thermal effects [138]. These are, among other things, highly dependent on the frequency and intensity of the acoustic field. The thermal effects are caused by the absorption of acoustic waves in the tissue and are the dominant factor of lethal implications in the MHz frequency range [138]. Non-thermal effects predominantly include cavitation, which involves the generation of gaseous bubbles by means of acoustic waves. In this case, mechanical damage can occur as the bubbles rapidly expand and shrink with the acoustic wave, causing high pressures and temperature changes [72]. Ultrasonic cavitation generally arises at frequencies that are substantially lower than 1 MHz, at least, for most tissues and when no bubble nuclei are already present. Moreover, lipid and aqueous zones, where bonding forces are low, are often initially prone to bubble formation [138].

#### 7.4.2. Air

Hearing loss is by far the most known bioeffect of sound waves at too high intensities. However, when it comes to ultrasound (US), adverse health effects are less clear. Presently, little is known about the dependence of various symptoms, such as headaches, fatigue, nausea, etc. on ultrasound [139]. Despite the disagreement on the effects on humans [139,140,141], there is still a consensus amongst independent organizations on the exposure limitations for airborne ultrasound [140]. The Health Canada report [142] provides an overview of the generally accepted exposure limits. For a clear overview of the full regulatory situation, standards, available information on environmental and health effects, and investigation of ultrasound interference effects, we refer the reader to [143].

### 7.5. Exposure to Laser Beams

LPT techniques use a high power density laser up to several kilowatts. Such high density burns anything that passes through the laser beam, e.g., vehicles, planes, birds. It is obvious that a safety system is needed, for example through a scanning ight detection and ranging (LIDAR). This shuts off the system as soon as an object approaches the beam. After the object has passed, the beam is turned on again [49]. Even at lower power, LPT can be dangerous and can cause serious eye damage since it uses infrared wavelengths that are in the retinal hazard region, beyond what ordinary human vision can see. A laser with a power of 3 mW can already cause eye damage. Burns may occur if more than 10 W is used [144]. This eye problem could be solved by using light with a longer wavelength, situated outside the retinal hazard region. However, LPT techniques using these wavelengths have lower efficiency and a much higher cost, and they do not solve the burning problem [49].

Particularly important standards are the IEC 60825-1 international laser safety standard of the International Electrotechnical Commission (IEC) [145], which is fully adopted by the European standardization organization as EN 60825-1 and has a classification ranging from class 1 to class 4, and the ANSI Z-136 standard [146]. Since LPT techniques are more efficient when using a pulse signal, flicker can arise which can also cause serious health problems when the frequency is too slow (headache, tiredness, decreased vision, an increased heart rate for people with anxiety disorder, an enlarged saccade, or even seizures) [147,148]. To avoid these problems, the IEEE 802.15.7 (the light communication standard) proposes a minimum safe modulation frequency of 200 Hz for light communication applications or other pulse-driven light applications [149].

## 8. Implementation and Operational Challenges

While different concepts for WPT offer an interesting potential, their implementation and operation in actual applications raises challenges. We here zoom in on those related to alignment, localization, and high power levels.

### 8.1. Alignment Challenges

In IPT and CPT system, alignment is imperative. Poor alignment results in low coupling factors and reduces the link efficiency, as was demonstrated in Section 3.2. For example, looking to the Qi specification, the coupling factor typically lies between the 0.3 to 0.6, thus lateral and angular misalignment can be tolerated to a limited extent. Higher misalignments are tolerated in MRC systems. These technologies can handle lower coupling factors between the two coils while maintaining relatively high link efficiencies. Nevertheless, this loosely coupled system has restrictions concerning the operation distance. The receiver still should be located in the reactive near field region of the transmitter. Non-coupled LPT systems such as WiCharge use the DLC architecture. Using a laser resonator in combination with retroreflectors provides a self-aligned system, as described in Section 3.2.

### 8.2. Localization Challenges

In uncoupled systems, the transmitter should preferably know the location of the receiver, as power can be steered in one or multiple specific directions. The location estimation must be accurate in the case of LPT, but may be less strict for RFPT. Further analysis of localization possibilities for these technologies is beyond the scope of this survey. Transferring energy on the basis of UVs comes also requires localization of the energy-constrained devices or IoT nodes. The UV should navigate to the nodes by using its onboard global navigation satellite system (GNSS) system. It is insufficient to rely solely on the GNSS system since the accuracy is limited to about 2 m in open areas and 5 m in forested landscapes. More precise positioning and alignment can be done using passive systems such as a camera on the UV and markers on the node. On the contrary, an active system based on, for example, sound waves can be used with the advantage of having a longer detection range compared to the passive option. The clear disadvantage is that the node always needs an amount of remaining energy to capture or transmit sound waves. Alternatively, an energy transfer link based on an uncoupled RF system can provide just enough energy for the localization process. After localizing the node, the battery can be recharged by relying on, for example, an inductive link [104].

### 8.3. Challenges at High Power

WPT is becoming more integrated into our society, just think of the wireless chargers for mobile devices such as mobile phones, laptops, and watches. It operates with power levels ranging from several watts to 100 W. In the case of higher power levels, more parameters must be taken into account to guarantee safe and efficient power transfer, which inhibits the breakthrough of large power WPT.

#### 8.3.1. Standardization

There is currently no high power mature equivalent for the Qi specification. Every manufacturer creates a proprietary architecture that is not compatible with other systems. Typically, interoperability is one of the strengths of wireless technologies. Only standards for the automotive sector are available, as discussed in Section 6. This explains the slower development of high power WPT systems as the manufacturers must develop a complete system rather than just the transmitter or receiver.

#### 8.3.2. Electromagnetic Compatibility

The European Union introduced Directive (2014/30/EU) on electromagnetic compatibility, where emc is defined as follows: *“Electromagnetic compatibility means the ability of equipment to function satisfactorily in its electromagnetic environment without introducing intolerable electromagnetic disturbances to other equipment in that environment.”* [150]. This directive limits the electromagnetic interference (EMI) from electrical appliances so that, when properly used, they do not interfere with radio and telecommunications systems. The EMC standards are divided into immunity and emission standards.

Larger power transfers go hand in hand with stronger fields, and of course, higher currents and voltages are present. The inverter at the transmitter side will therefore be a larger source of electrical-noise pollution for several reasons:The amount of transferred power can be controlled by adjusting the duty cycle of the PWM signal in the inverter. This change can result in the loss of zero voltage switching off the power switch and cause high voltage changes in time $\frac{dv}{dt}$. Subsequently, a changing magnetic and electric field is created, which carries the high $\frac{dv}{dt}$. The fields around conducting components cause common-mode currents to flow from the system to the environment and back via the mains. This results in **conducted interference** or, more precisely, **common-mode interference**.**Radiated EMI** can be induced by switching large currents in the inverter, which cause large current changes in time $\frac{di}{dt}$. Furthermore, radiated EMI can also be caused by the leakage field of the inductors, due to poor coupling between transmitter and receiver.

#### 8.3.3. Heat Dissipation at High Power Wireless Charging

Most of the high-power wireless charging systems based on inductive coupling have a system efficiency between 80 and 90%. In a 5 kW system, these efficiencies correspond to a heat loss of approximately 750 W. These losses will rise system temperature, which has an effect on the efficiency [151]. Power losses in high-power charging systems can be caused by the properties of those systems. The main causes for heat losses are Joule heating, skin effect, proximity effect, switching losses in the power stage, and rectifier losses at the receiver.

**Joule heating** is a phenomenon that occurs when a current flows through a cable or component. Each cable, connection, path, etc. has a certain resistance, which in combination with the current causes a power loss, $P=RI^{2}$.

**Skin effect** occurs when an alternating current flows through conductors. This phenomenon causes the current to flow through the "skin" of the conductor. The higher the frequency, the smaller the skin depth, which results in a higher resistance. This in turn will cause the cable to heat up and cause extra joule losses. Hence, a snowball effect can occur: warmer cable sheathing can increase the resistance of the cable, more heating, etc. The skin depth can be calculated by:(60)$$d=\sqrt{\frac{2}{\omega \;\cdot \;\mu \;\cdot \;\gamma }}$$
where ω is the angular frequency of the alternating current, 2π·frequency (rad/s), μ is the magnetic permeability of the conductor (H/m), and γ is the resistance of the conductor (S/m).

**Proximity effect** refers to a phenomenon that occurs when two conductors, in which an AC current flows, are close to each other. This will result in Eddy currents and change the current density. When the currents in adjacent conductors have the same direction, the current will concentrate on the outside of the conductor. Vice versa, when the two currents are opposite, the current will concentrate on the inside of the conductor. The Eddy currents in proximity effects are created due to the variable magnetic field of the current in the adjacent winding layer, whereby the amplitude of the Eddy currents increases exponentially with the number of coil windings/layers [152].

**Switching losses in the power stage:** Switching components, such as MOSFETs, need a certain time to switch on and off, which causes losses. This reduces the efficiency of the power stage. The switch heats up faster, which may have other consequences, such as a higher internal resistance or in the worst case exceeding maximum temperatures.

**Rectifier losses at receiver side:** Power can be transmitted using high-frequency currents. These are usually not directly usable in typical applications, so currents must be rectified at the receiver side. In general, a passive rectifier with diodes is used. When a diode conducts, there will be a forward voltage over it. This will cause high power losses because high currents will flow.

## 9. Use Cases—Technology Mapping Catalog

Following the conceptual and engineering view on WPT technologies, an overview of potential use cases in different environments is provided below. The most promising candidate technologies are mapped to the applications according to required power levels, which are summarized in Table 4.

### 9.1. Living and Working Environments

Recharging and operating appliances without the need for wires is convenient or even preferable for many devices in smart homes and living environments. WPT offers the opportunity to reduce supererogatory cables and also avoids connection issues caused by dirt or broken connectors. The distribution of energy within domestic environments could be handled in a different way, providing flexibility in the transition of life stages. For example, it may be easier to convert a nursery into office space. WPT can be applied to a huge field of applications, ranging from easily movable baby monitors to cordless rechargeable wheelchairs. It is envisioned that wireless chargeable appliances could create an environment that is more user-friendly and provide opportunities for independent aging.

WPT can be used in different power classes. In smart home environments, there are many relatively low-energy consumable applications, such as chargeable smartphones, wireless speakers, and wireless light switches. On the other side, similar WPT technologies are also suitable for high-energy consumable appliances such as wireless coffee makers, wireless work tools, and wireless televisions. The even higher power deliveries are intended for energy transmission between the grid and the household vehicles such as the EVs, electric scooters, or ride-on mowers. A survey covering this scope with their proposed suitable WPT techniques is given in Table 4. The applications are divided into different classes, based on their nominal power usage. Notice that short-term peak powers, e.g., during start-up, can be multiple times higher. The previously discussed relevant standards are mentioned, regardless of whether these were developed for this purpose. Thus, this table does not imply that the standards within a certain power category are suitable for all applications. In addition, we assume that the listed power categories are minimally required to ensure the proper operation of the device. Some batteryless devices have strict power delivery requirements, otherwise, quality may be compromised.

A brief comparative analysis can allow the reader to make the appropriate choice for a specific application. It is mostly recommended to use inductive coupling from low to high power applications. The WPC standards offer good solutions for a wide range of power needs. Unfortunately, three of the four standards are still under development. The Ki is already available to members of the WPC, yet only the Qi standard is openly available. The counterpart of WPC called Airfuel offers the standard Airfuel RF described in the IEC63028 standard, yet is only available to members. This MRC technology offers power delivery up to 50 W and is usable in applications where more flexibility in the form of spatial freedom is required. CPT also offers good properties in terms of power transfer possibilities, yet there is no international standard and only a few commercial applications are available. The ISO, IEC, and SAE standards are included in the table and show that for these high power links, standards already exist. Unfortunately, mid-power standards are still missing, although WPC intends to close this gap. The uncoupled technologies such as LPT have the opportunity to cover a wide range of applications with different power consumption rates. Unfortunately, this is currently not often implemented because standards are not yet available. Additional safety measures should be considered when high light beams or RF energy travel through the air. If the ISM band limits are not exceeded and the laser power class remains under class 1, no specific safety measures are legally required. Thus, in residential environments, LPT and RFPT can serve low consumable applications.

### 9.2. Environmental Monitoring, Industry 4.0, and Logistics

A quickly customizable workspace is desirable within Industry 4.0 and can be met thanks to wireless power transfer. For example, a production line can easily be extended or adapted with new devices, or the factory floor, in general, can be rapidly reconfigured. This enables faster operation times, which can increase production efficiency and results in lower production costs. In addition, it offers advantages in places where cabling is difficult. This makes operations such as drilling or welding underwater, in very hot or chemical environments easier. Energy can also be supplied to, for example, rotating elements within machines. Furthermore, WPT technologies ensure continuity of workability for moving objects, such as forklifts, robots, UVs, drones, etc. In normal situations, the vehicle battery is recharged by means of a docking station, resulting in a stationary and therefore unusable device. WPT in industrial environments could provide energy to moving objects, allowing them to be used continuously. Moreover, these environments often contain a lot of dust or other dirt particles that can creep into equipment. The equipment could be better designed and sealed. This could extend the lifetime of the device and shorten the downtime. Adding WPT to new industrial applications or to already existing equipment, in a non-invasive manner, is feasible and will create more efficient work equipment. Some applications are listed in Table 4, with their associated standards, technologies, and power classifications. WPT applied in Industry 4.0 should enable both wireless communication and energy to devices, thus reducing cabling issues. Furthermore, AGVs can operate all day long, or labels for goods in warehouses can harvest their energy continuously out of the environment. The latter has the advantage that prices can be adjusted more quickly since both communication and power are wireless. It can make the business flow more efficient. Additionally, the internal batteries no longer need to be replaced. WPT can also easily charge products, such as consumer electronics, that already have to be sold partially charged, resulting in a reduction of the operators’ workload.

Most industrial applications have high power consumption, thus making IPT and CPT the most interesting technology due to the high energy densities. Certainly, it is important to take the previously discussed safety factors into account and to consider the challenges related to high energy transfer. Table 4 lists a number of devices that can benefit from a built-in WPT system. New emerging standards of the WPC, such as the LEV standard and the Industry standard, fit perfectly for these use cases. Additionally, low-power applications are also retrievable in the industry, such as monitoring applications that obtain a lot of data and maintenance information with the help of distributed low-power sensors. Similarly to smart home applications, RF energy transfer is an appropriate technology to power all these devices. If sufficient research is provided, the more expensive LPT could be considered for future applications, especially for large industrial sites, where only the LPT approach could provide wireless energy over high distances. For example, powering a water pump from a reservoir located far outside the factory.

## 10. Current Gap and Future Trends

Through different WPT techniques, it is feasible to deliver MWs over short distances and significantly lower power levels over longer distances. Charging over large distances with high power is feasible but challenging, especially if the system should comply with the regulations discussed in Section 7. Moreover, making it efficient is even more difficult. Novel techniques are being proposed to address this efficiency degradation over large distances, while still maintaining the regulations. Although this has its limits. The current standards and regulations are not adapted to the required power levels to energize further located devices. This section summarizes the gaps and trends of the individual topologies mentioned in Section 2, Section 3, Section 4.

### 10.1. Electromagnetic Coupled: Gains and Trends

Physically speaking, the highest energy transfer efficiency can be achieved with IPT and CPT systems, since the link efficiency can be close to 100 percent with very closely spaced coils or capacitors respectively. By using smart techniques for generating the amplified AC voltage, working with soft switching circuits, and reducing the rectifier losses, the overall efficiency of IPT and CPT systems, compared to a wired connection, can achieve similar efficiencies. However, it remains a challenge to keep this system affordable, as a wired solution mostly has a cheaper bill of materials (BOM) cost. Moreover, the transmitter and receiver have to stay close together, which limits the spatial freedom and flexibility. The maximum achievable link efficiency in loosely coupled systems physically has its limits. In order to keep the overall efficiency high, more efficient active rectifiers for electromagnetic coupled systems are being investigated, e.g., in [153,154]. We also see a trend of shifting from hard switching to soft switching with ZVS design resulting in lower losses in the FETs. The wide band gap FETs is considered in new designs, enabling lower losses at higher frequencies compared to the conventional MOSFETs. Efficient high-frequency converters built with these newer FETs allow e.g., the usage of low-cost PCB coils instead of using the more expensive Litz wire coils. Several years ago, studies were started on magnetic beamforming and magnetic MIMO to increase range and efficiency. Kisseleff et al. [97] show an efficiency gain of 37% compared to conventional solutions. The SoftCharge concept in [155] and MagMIMO in [94] operate on the basis of magnetic resonance coupling and shows the potential to further increase spatial freedom compared to traditional MRC implementations. It is rather unclear whether these approaches have future perspectives in consumer applications, as currently only the Qi standard is adopted by the appliance community. The MIMO concept is being further explored for capacitive systems. Current models consist only of ideal capacitors. Therefore, further research is needed into the losses in the coupler. This will give a better estimation of coupling and system efficiency [107,156,157]. Technologies and developments for charging EVs and robots are another important area of research, such as dynamic wireless power transfer (DWPT) and multiple phase WPT [158]. To make DWPT practically affordable, e.g., Inoue et al. [159] present a technique to use a single inverter connected to multiple transmitter coils. Moreover, CPT is proposed for DWPT due to its lower cost, in terms of manufacturing the coupler, and ability to transfer energy, without losing efficiency, in the vicinity of metal object [160,161,162].

### 10.2. Electromagnetic Uncoupled: Gains and Trends

Over the last decade, we have seen an emerging trend in uncoupled wireless power transfer by means of RF, light, and acoustic waves. The power density at the receiver is often low due to path loss, yet their ubiquity provides a huge advantage in terms of distance and receiver location. Several techniques have been identified in recent research that may ensure further improvement of the RF power transfer efficiency. (I) Single [85,163] and distributed antenna arrays [87,164,165] have been introduced to compensate for the high path loss, either by increasing the array gain, lowering the distance between the transmitter and receiver, or a combination of both [166]. This, however, comes at the expense of multiple transmitters and increased system complexity, and thus costs. Transmission and safety regulations pose a limit on the radiated power, which consequently caps the achievable gain. Moreover, in the current regulations, transmission power limitations are often not a function of the antenna setup, giving currently little to no benefits to directional systems over omnidirectional transmission. Next to conventional far-field beamforming, dynamic focusing using large arrays has been introduced to form focal points that maximize power transfer [86]. (II) Since the radiated power is ultimately bound by regulation, a lot of effort is invested in increasing the efficiency of the receiver rectifier [167]. However, the low voltages available at the rectifying diodes/transistors make it challenging to limit conduction losses, hence the rectifier efficiency is low at low input power. Moreover, the input power is variable and unpredictable as it depends on many practical circumstances, such as distance, obstacles, etc. Therefore, a lot of research is focused on improving the efficiency of rectifiers over a wide input power range [168]. In addition, a shift towards higher frequencies [169] and broadband rectifiers [170,171] for multiband energy harvesting brings continuous change to the rectifier landscape. (III) It is expected that swipt will bring many opportunities to applications in IoT. While it allows for efficient use of the RF spectrum, additional challenges are imposed on rectenna design. Not only hardware but also applying an optimal signaling waveform, such as peak-to-average power ratio (PAPR) signals, has proven to impact rectifier design significantly [172].

Section 3.2 indicated that the current laser power transfer systems overall efficiency achieves values below 15%. Although these systems have better efficiency performance than RFPT, there are still efficiency gains to be made in the electrical-to-optical and optical-to-electrical conversions. DLC is another OWPT technique and comes with advantages such as the self-aligning feature and the opportunity to charge multiple receivers with one transmitter. These systems are capable of achieving efficiencies of up to 25% thus making this technology interesting for future IoT applications. This technology can be extended by combining power transfer with communication over optical power. SWIPT via DLC can therefore be further explored, since there is a huge bandwidth available [64,65]. Similar to beamforming in magnetic and RF based-systems, optical phased arrays (OPA), to form a focal point in 3D, can be further explored [86].

Acoustic power transfer arises in applications where device miniaturization is important or em-based power transfer due to metal shielding is difficult. However, the technology comes with a number of challenges and drawbacks, with the result that research is far more limited than all other fields. For example, acoustic waves are considerably more subjected to air absorption than electromagnetic waves. Moreover, the transmitted acoustic power is rather limited in both biomedical and air applications due to exposure limits, as covered in Section 7.4. Similar to the aforementioned WPT technologies, transducer arrays have been introduced in acoustic WPT systems to improve the overall power conversion efficiency [93,173].

## 11. Conclusions

Many applications can and could benefit from wireless charging. This enhances convenience in not needing cables and reducing vulnerabilities introduced by contacts. More disruptively, it enables flexible living and working environments and opens opportunities for new applications in, e.g., environmental monitoring. WPT technologies can hence also contribute to societal challenges such as reducing e-waste from batteries and supporting more flexible and eventual circular use of equipment and buildings. In this paper, a survey of different technological candidates for wireless power transfer was given. We have treated them at a conceptual level, and also detailed typical transmit and receive circuits. Basic equations clarify the achievable efficiency. We can conclude that high efficiencies are feasible at small distances, yet higher operating ranges can be achieved with uncoupled technologies with the drawback of suffering from an efficiency drop. Acoustic power transfer is the only technology that is not EM-based, making it suitable for places where no EM waves are allowed. Additionally, the fact that acoustic signals penetrate better through metal walls compared to the other discussed WPT technologies, makes this a suitable solution in several situations.

Innovative ideas were presented to improve efficiency, range, and achievable power levels in WPT. Additionally, important engineering aspects in making wireless power transfer technologies a reality for a variety of applications were covered. These range from standardization and safety measures to design challenges. The combined insights result in an overview of envisioned applications, and corresponding candidate technologies.

A critical remark should be made regarding energy efficiency. It is clear that wireless charging technologies come with a penalty with respect to their wired counterparts. The higher the power one wants to transfer and the larger the distance to be bridged, the more considerable the losses are. In view of challenges related to climate change, one can not neglect that. Efficiency should hence be critically assessed and optimized, in particular for higher energy-consuming appliances. WPT systems can also be considered in the broader context of the energy transformation. Indeed, rechargeable energy storage could be opportunistically recharged when energy from renewable sources is available. Larger batteries e.g., in electrical vehicles, could contribute to a better alignment of self-production and self-consumption in home environments, with little effort thanks to cordless charging.

Finally, it should be acknowledged that an overview is never complete and is outdated from the moment it is written. Interesting new approaches to improve WPT are being researched.

## Figures and Tables

**Figure 1 sensors-22-05573-f001:**
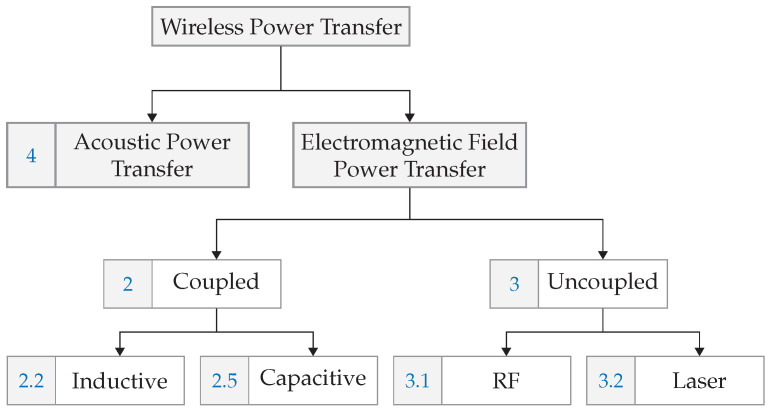
Categories of wireless power transfer systems, including section numbers.

**Figure 2 sensors-22-05573-f002:**
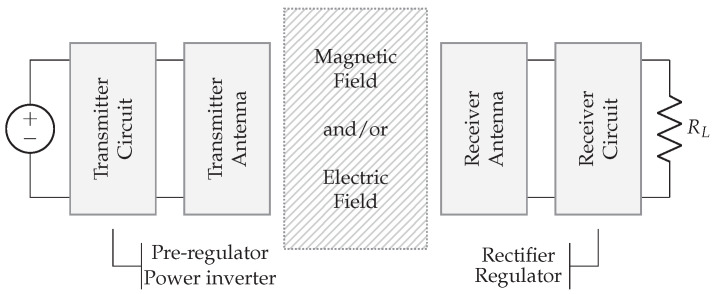
Structure of a coupled wireless power transfer system.

**Figure 3 sensors-22-05573-f003:**
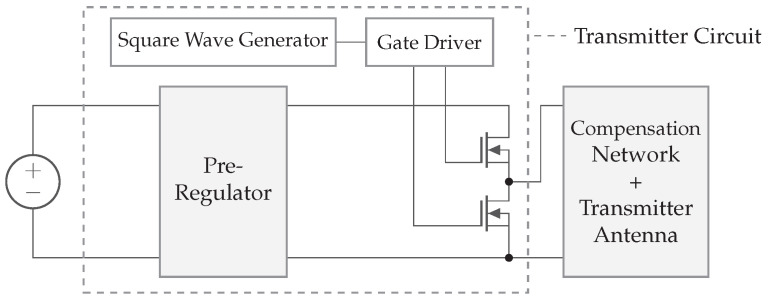
Transmitter example with pre-regulator and half-bridge inverter.

**Figure 4 sensors-22-05573-f004:**
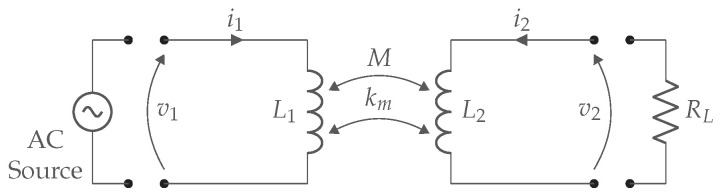
Basic circuit of an inductively coupled WPT system.

**Figure 5 sensors-22-05573-f005:**
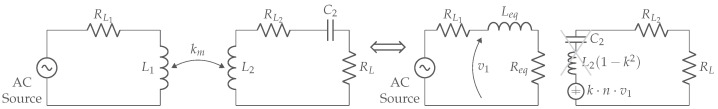
Canceling of the secondary leak inductance. The figure is composed of the equivalent model with a voltage-controlled secondary voltage source, with *n* being the inductance ratio [6].

**Figure 6 sensors-22-05573-f006:**
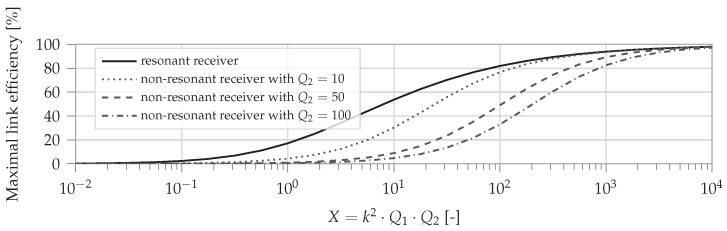
The maximal efficiency for a series resonant compared to a non-resonant secondary circuit as a function of the coil quality factors and the coupling factor with $a=a_{max}$ [6].

**Figure 7 sensors-22-05573-f007:**
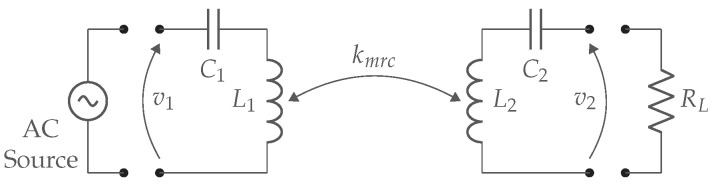
A series tuned primary and secondary LC-tank. The coupling factor $k_{mrc}$ in a magnetic resonance coupling system is typically lower than a coupling factor $k_{m}$ in an IPT system.

**Figure 8 sensors-22-05573-f008:**
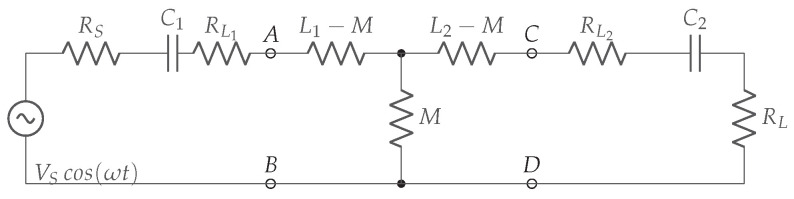
MRC system with one receiver: equivalent scheme.

**Figure 9 sensors-22-05573-f009:**
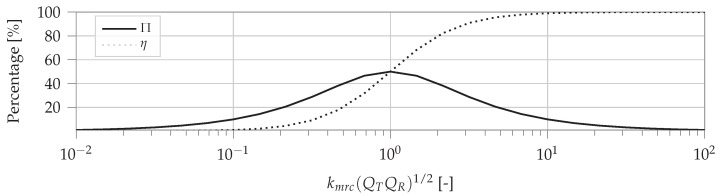
The transfer coefficient Π and efficiency η as function of $k_{mrc}{\left(Q_{T}Q_{R}\right)}^{1/2}$ [11].

**Figure 10 sensors-22-05573-f010:**
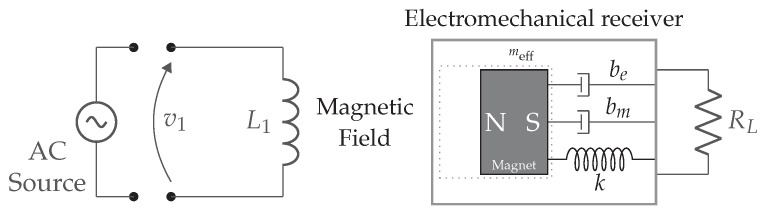
Electrodynamically coupled WPT system with electromechanical receiver [13].

**Figure 11 sensors-22-05573-f011:**
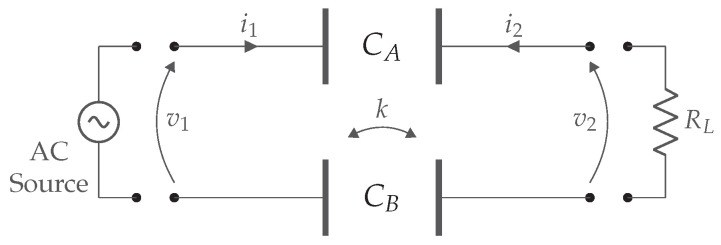
Basic circuit of a capacitive coupled WPT system.

**Figure 12 sensors-22-05573-f012:**
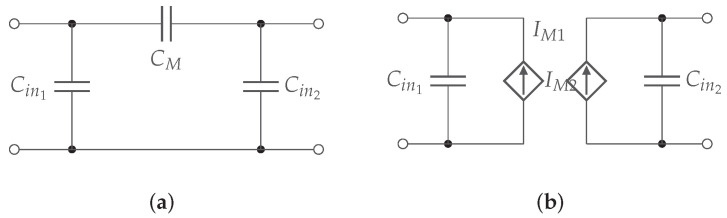
Equivalent circuit models for the capacitive coupler. (**a**) Pi-model, (**b**) equivalent model with voltage controlled current sources with $I_{M1}=j\omega C_{M}V_{C_{2}}$ and $I_{M2}=j\omega C_{M}V_{C_{1}}$.

**Figure 13 sensors-22-05573-f013:**
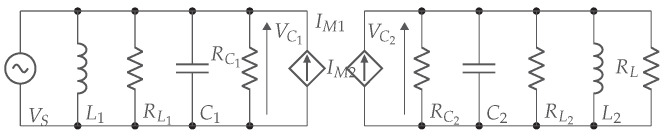
Equivalent scheme with VCCS model.

**Figure 14 sensors-22-05573-f014:**
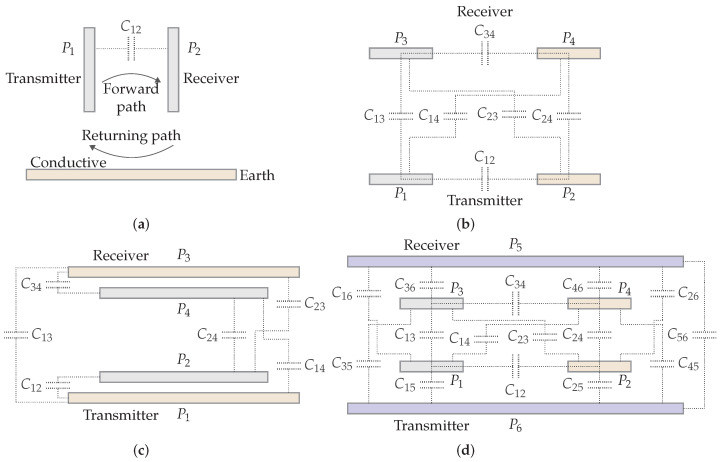
CPT structures: (**a**) two-plate, (**b**) four-plate parallel, (**c**) four-plate stacked, and (**d**) six-plate structure.

**Figure 15 sensors-22-05573-f015:**
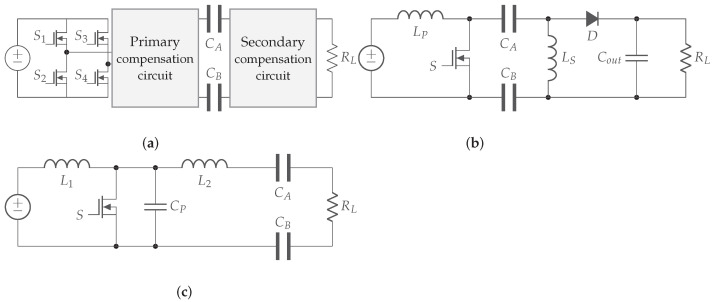
Circuit topologies for the realization of CPT systems. (**a**) General full-bridge inverter. (**b**) Modified PWM-based sepic converter. (**c**) Modified class-E converter.

**Figure 16 sensors-22-05573-f016:**
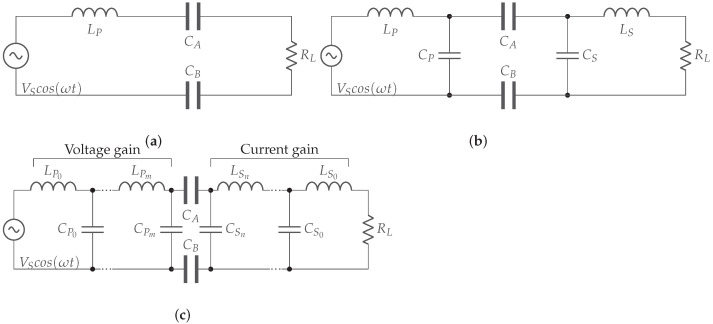
Compensation circuit topologies for practical CPT. (**a**) L-compensation circuit. (**b**) LC-compensation circuit. (**c**) Multistage L-section compensation circuit.

**Figure 17 sensors-22-05573-f017:**
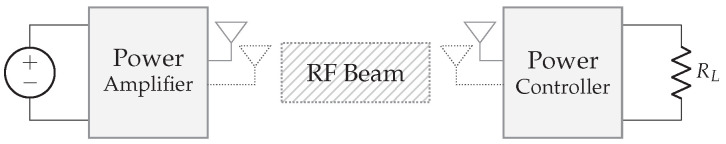
Schematic of an RFPT system. Antenna arrays (in dashed line) at the transmitter and/or receiver can be employed to increase the power transfer.

**Figure 18 sensors-22-05573-f018:**
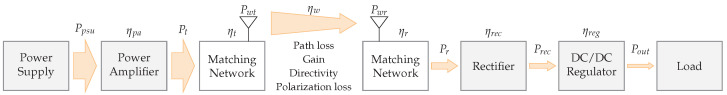
Overview of the system concept, transmission, and efficiency model for RF power transfer.

**Figure 19 sensors-22-05573-f019:**
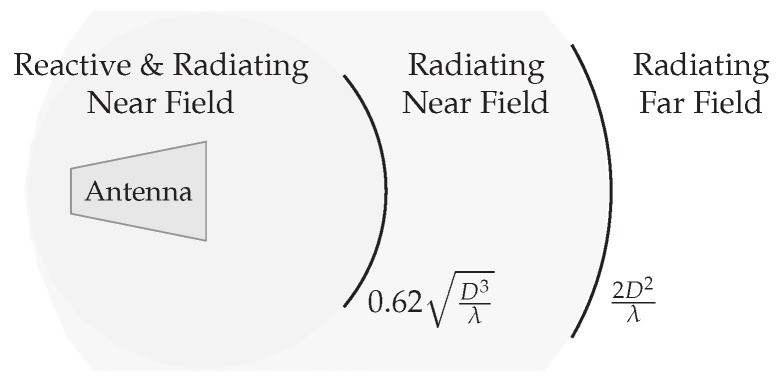
Three fields where the receiver can be located: reactive and radiating near field, radiating near, and far field. Distances depend on wavelength λ and antenna dimension D.

**Figure 20 sensors-22-05573-f020:**
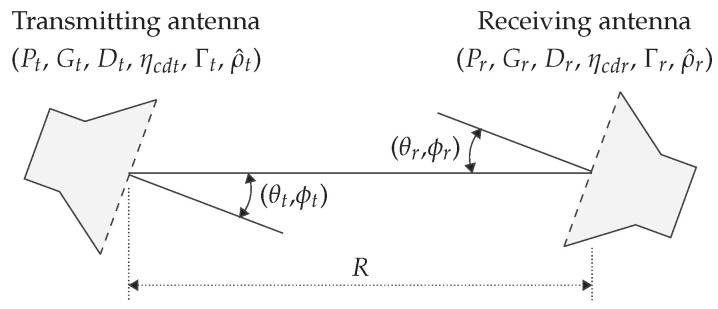
Orientation and parameters of the transmit and receive antennas, based on [46].

**Figure 21 sensors-22-05573-f021:**
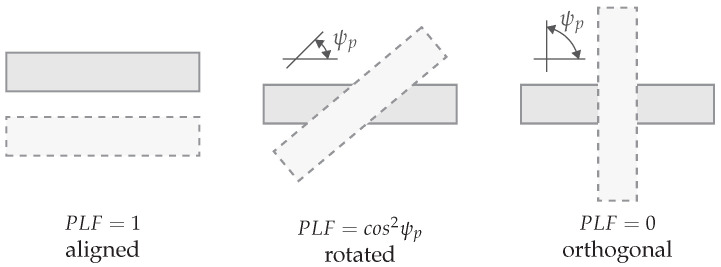
Transmitting and receiving antennas PLF visualization example (similar for linear wire antennas), based on [46].

**Figure 22 sensors-22-05573-f022:**
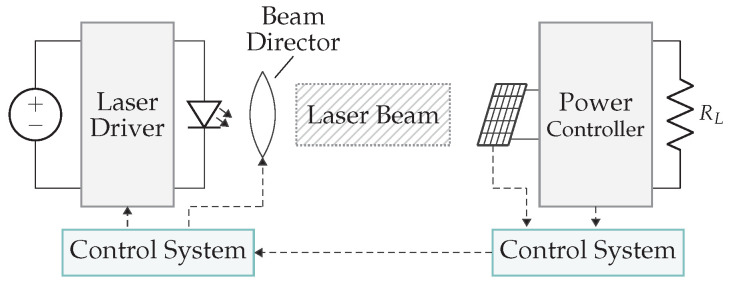
Schematic diagram of an HILPB system [49].

**Figure 23 sensors-22-05573-f023:**
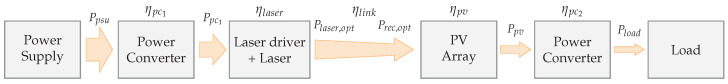
Efficiency of an LPT system.

**Figure 24 sensors-22-05573-f024:**
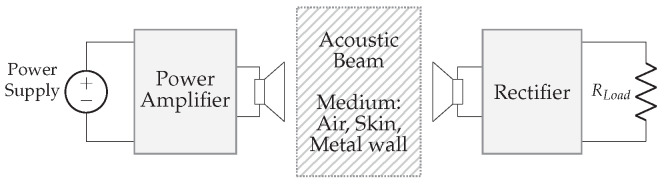
Basic structure of an APT system.

**Figure 25 sensors-22-05573-f025:**
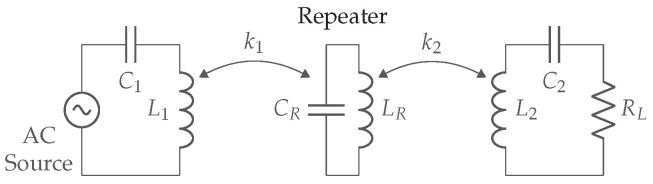
Example of magnetic resonance coupling with LC repeater.

**Figure 26 sensors-22-05573-f026:**
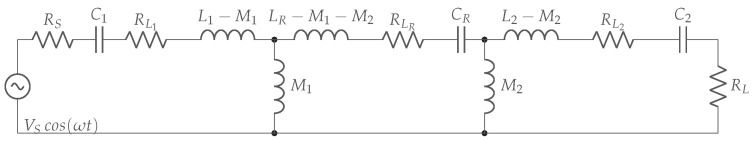
Equivalent scheme of an MRC system with LC repeater.

**Figure 27 sensors-22-05573-f027:**
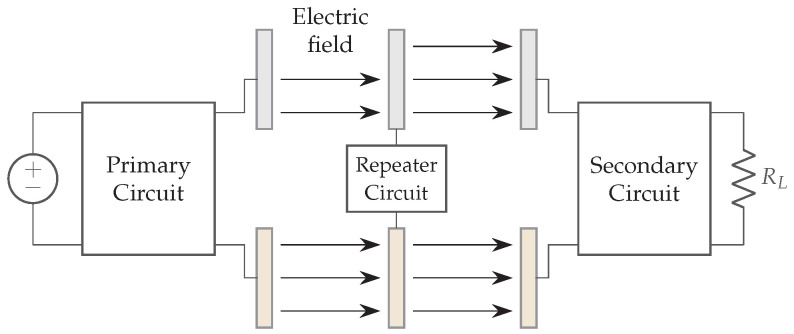
CPT system with one repeater.

**Figure 28 sensors-22-05573-f028:**
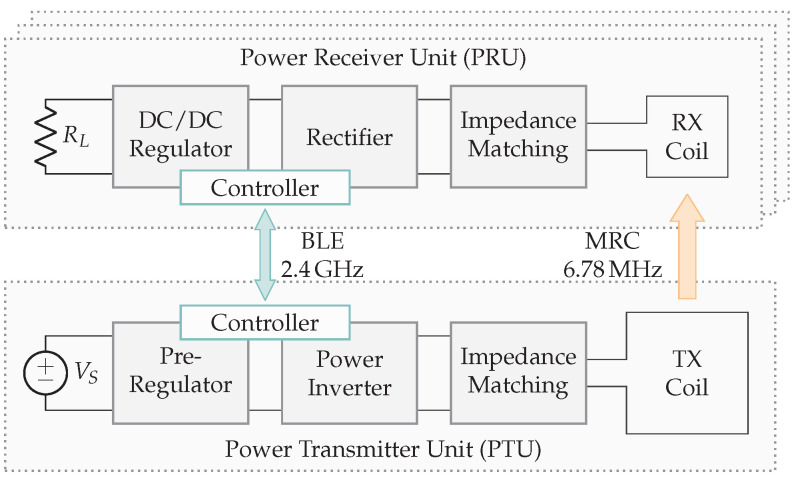
AirFuel Resonance system representation with one PTU and multiple PRUs [110].

**Figure 29 sensors-22-05573-f029:**
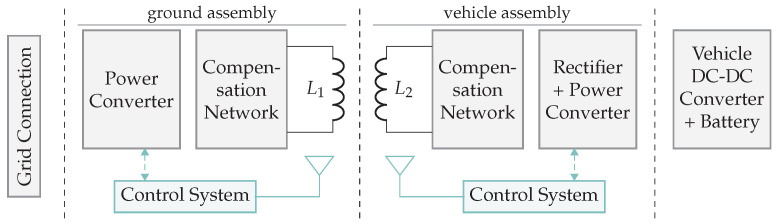
SAEJ2954 block diagram.

**Figure 30 sensors-22-05573-f030:**
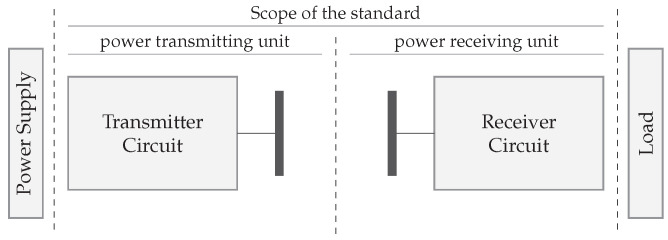
Scope of the ARIB STD-T113 standard [120].

**Figure 31 sensors-22-05573-f031:**
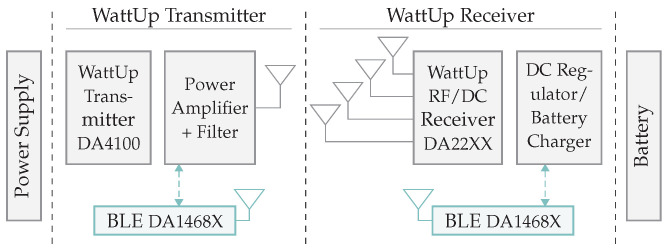
The near-field WattUp system consists of a Wattup-compatible transmitter and receiver. An optional BLE connection enables communication between both, for example, communicating the battery status information [124].

**Table 1 sensors-22-05573-t001:** Baseline characteristics of different wireless power transfer technologies.

WPT Technology		Power Transfer	Range	Frequency	Efficiency	Biological Impact
Inductive	IPT	W to MW	cm	kHz to MHz	high	minor
Capacitive	CPT	W to kW	mm/cm	kHz to MHz	high	minor
Laser	LPT	W/kW	m/km	>THz	medium	medium/significant
Radio frequency	RFPT	mW/kW	m/km	MHz to GHz	low	medium/significant
Acoustic	APT	mW/kW	m/cm	kHz to MHz	medium	significant/medium

**Table 2 sensors-22-05573-t002:** Summary of a selection of LPT realizations in the literature.

Laser Type	Beam Output Power (W)	Wavelength (nm)	PV Type	Distance (m)	Received Usable Power (W)	Overall Efficiency (%)	Ref.
Nd:YAG laser	5	523	InGaP	200	<1.75	<14	[56]
Diode	1500	940	Si	15	7	<8.5	[57]
Diode	60			1000	<10	<10	[58]
Diode	300	808	GaAs	50	40	<14	[58]
Diode	360	808	GaAs	50	90	<14	[59]
Diode	25	793	GaAs	100	9.7	11.6	[60]
Nd:YAG laser	160	1064	Si	3	19	<14	[61]
Diode	0.05	661	Si	0–4	<$8\times 10^{-3}$	0.5–4	[62]

**Table 3 sensors-22-05573-t003:** Comparison of materials and their influence on the resulting capacity of the CPT system [108].

Material	Relative Permittivity $\varepsilon_{R}$	Capacitance (pF)
Air	1.0005	1.77
Rubber	3.0000	3.81
pvc	4.0000	5.31
Glass	7.6000	13.45
Water	80.1030	141.78

**Table 4 sensors-22-05573-t004:** Available and emerging standards and technologies for residential and industrial applications.

Power Range	Adequate Technologies	Appropriate Standards	Currently Available	Applications
Industrial	Residential
<10 W	IPT, MRC, CPT, LPT, or RF based	WLC	✔	Sensor nodes, IoT devices, remote controllers, smoke detectors	Switches, remote controllers, wearables, keyboards, clocsk, alarm systems, electric toothbrushes, LED lights
AirFuel RF and Resonant	✔
Qi standard	✔
10–100 W	IPT, MRC, CPT, or LPT	Qi standard	✔	Work tools, small displays, mobile devices	Mobile devices, monitors, small robots, speakers, electric curtains, computers, televisions, consoles
KI Cordless Kitchen	
AirFuel Resonant	✔
100–450 W	IPT, CPT, or LPT	Medium Power standard		Small machines or tools, lightweight drones or UVs, lightweight robot arms	E-steps, e-bikes, push lawn mowers, mobility scooters, electric wheelchairs
KI Cordless Kitchen	
LEV	
Industry standard	
850 W to 2 kW	IPT CPT, or LPT	KI Cordless Kitchen		Big drones or UVs, robot arms	Mixers, shutters, cookers, vacuum cleaners, hair dryers, drills, hedge trimmers, washing machines, electric heaters, lawn tractors, electric cars
Industry standard	
ISO 19363	✔
IEC 61980	✔
SAE J2954	✔
Tens of kW	IPT	Industry standard		Heavy-duty UVs, heavy-duty electric vehicles, bigger machines	Heavy duty electric vehicles
SAE J2954/2

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
