# Peer review of "Wireless Power Transfer: Systems, Circuits, Standards, and Use Cases"

_sensors, 2022, doi:10.3390/s22155573_

Round 1

Reviewer 1 Report

This paper reviews WPT technology from many aspects from system types, topology, standard and applications in a good organization. It covers almost every part of the technique. The contents and discussions are reasonable.

The paper is well written, however, the paper is too long. It is hard to follow a paper for almost 50 pages. I suggest the author to make it more concise. Also, a simple discussion concluding the general gaps, concerns and future orientations can be included. Overall, it is a very good review paper for those who wants a whole picture of the technique.

Author Response

Please find the point-by-point responses in attachment.

Reviewer 2 Report

The paper is about a very interesting and actual subject. The paper is well written and brings an extensive overview of the existing techniques for Wireless Power Transfer (WPT), presenting figures and a critical analysis of each topology, highlighting the strengths and weaknesses of each one. The main technologies, manufacturers, standards safety issues and standards are presented.

The extensive amount of information, presented in a concise and well-placed manner, could serve as an interesting research source for researchers looking for information on Wireless Power Transfer (WPT).

The article brings an overview of the main technologies used in the WPT, with a critical view of each one. The article does not present any new information to the area, however, it can be an interesting source of research, as a starting point for understanding such technologies. In this way, the main questions addressed by the research are being able to place in a single article, aspects related to the main technologies of WPT.

As specified earlier, the article does not address an original topic per se, nor does it address a gap in the area. It addresses, however, aspects that can be considered relevant to the area, since points about the main WPT technologies can be found in a single article.

In relation to the subjects found in other published materials, this article does not present any point of novelty, it only aggregates in a single article, the authors' view in relation to other articles and sources already published in the area.

The purpose of the article is not to present a proposal for a new methodology. In this sense, there are no improvements to be made. However, other improvements can still be considered in relation to the article (please see comments regarding the references).

Considering that the article presents an overview of the main technologies in WPT, the conclusions are consistent with the proposal of the article.

Regarding the references, the article cites 159 references, among which 52 are from the last 5 years and only 21 from the last 3 years. In this aspect, as the main purpose of the article is to present a bibliographic review on WPT, the article could bring a greater number of more recent references, considering the progress obtained in the last 3 years.

Author Response

Please find the point-by-point responses in the attachment.

Reviewer 3 Report

In the row 125 and 147  the  title is the same -  "Receiver Circuit" -  the authors need to do correction  according to the description.

The paper is traceable and extensive. It is on the conceptual level, and  it is an overview of various technological candidates for wireless energy transmission is presented.

A significant contribution to the work would be if the authors gave a simple overview of the improvement of individual solutions compared to the existing ones (e.g., for the percentage that improves energy efficiency or by how much the losses in transmission are reduced...).

The transmission of higher power (MW) over longer distances was not given as much attention by the authors as to transmission over shorter distances and smaller amounts of power. Namely, they did not analyze , in the paper, what needs to be done in the transmission of larger  power wirelessly while reducing losses.

 In the conclusion, they should indicate what they recommend for further research.

Author Response

(The authors gave the same response as above.)
